# Endothelial heterogeneity across distinct vascular beds during homeostasis and inflammation

**Ankit Jambusaria[1,2], Zhigang Hong[1], Lianghui Zhang[1], Shubhi Srivastava[1], Arundhati Jana[3], Peter T Toth[1,4], Yang Dai[2], Asrar B Malik[1]\*, Jalees Rehman[1,3]\***

[1]Department of Pharmacology, The University of Illinois College of Medicine, Chicago, United States; [2]Department of Bioengineering, The University of Illinois College of Engineering and College of Medicine, Chicago, United States; [3]Division of Cardiology, Department of Medicine, The University of Illinois College of Medicine, Chicago, United States; [4]Research Resources Center, University of Illinois, Chicago, United States

**Abstract** Blood vessels are lined by endothelial cells engaged in distinct organ-specific functions but little is known about their characteristic gene expression profiles. RNA-Sequencing of the brain, lung, and heart endothelial translatome identified specific pathways, transporters and cell-surface markers expressed in the endothelium of each organ, which can be visualized at http://www.rehmanlab.org/ribo. We found that endothelial cells express genes typically found in the surrounding tissues such as synaptic vesicle genes in the brain endothelium and cardiac contractile genes in the heart endothelium. Complementary analysis of endothelial single cell RNA-Seq data identified the molecular signatures shared across the endothelial translatome and single cell transcriptomes. The tissue-specific heterogeneity of the endothelium is maintained during systemic in vivo inflammatory injury as evidenced by the distinct responses to inflammatory stimulation. Our study defines endothelial heterogeneity and plasticity and provides a molecular framework to understand organ-specific vascular disease mechanisms and therapeutic targeting of individual vascular beds.

**\*For correspondence:**
abmalik@uic.edu (ABM);
jalees@uic.edu (JR)

**Competing interests:** The authors declare that no competing interests exist.

## Introduction

Endothelial cells (ECs) line blood vessels in all tissues and organs, and they form a barrier which tightly regulates the trafficking of oxygen, metabolites, small molecules and immune cells into the respective tissue (*Liao, 2013*). Previous studies have suggested that the morphology of the microvascular endothelium or the expression of selected genes can vary when comparing the vasculature of multiple tissues, thus allowing ECs to take on tissue-specific EC functions (*Chi et al., 2003*; *Potente and Mäkinen, 2017*; *Aird et al., 1997*). Environmental signals from the tissue microenvironment including mechanical forces, metabolism, cell-matrix, cell-cell interactions, organotypic growth factors likely play an important role in regulating this endothelial heterogeneity (*Potente and Mäkinen, 2017*).

The tissue-specific interaction between ECs and surrounding cells occurs as early as during development, when, for example, brain ECs instruct neuronal differentiation (*Bussmann et al., 2011*; *Matsuoka et al., 2017*). Such tissue-specific endothelial adaptations persist throughout adulthood when brain ECs form a highly selective barrier composed of specialized tight junctions to limit neurotoxicity (*Pozhilenkova et al., 2017*). In the lung, ECs differentiate in parallel with epithelial cells to form gas exchange units which are in contact with the external environment and thus need to ensure a rapid immune response (*Jambusaria et al., 2018*; *Rafii et al., 2016*). Heart ECs, on the other

**eLife digest** Blood vessels supply nutrients, oxygen and other key molecules to all of the organs in the body. Cells lining the blood vessels, called endothelial cells, regulate which molecules pass from the blood to the organs they supply. For example, brain endothelial cells prevent toxic molecules from getting into the brain, and lung endothelial cells allow immune cells into the lungs to fight off bacteria or viruses.

Determining which genes are switched on in the endothelial cells of major organs might allow scientists to determine what endothelial cells do in the brain, heart, and lung, and how they differ; or help scientists deliver drugs to a particular organ. If endothelial cells from different organs switch on different groups of genes, each of these groups of genes can be thought of as a 'genetic signature' that identifies endothelial cells from a specific organ.

Now, Jambusaria et al. show that brain, heart, and lung endothelial cells have distinct genetic signatures. The experiments used mice that had been genetically modified to have tags on their endothelial cells. These tags made it possible to isolate RNA – a molecule similar to DNA that contains the information about which genes are active – from endothelial cells without separating the cells from their tissue of origin. Next, RNA from endothelial cells in the heart, brain and lung was sequenced and analyzed.

The results show that each endothelial cell type has a distinct genetic signature under normal conditions and infection-like conditions. Unexpectedly, the experiments also showed that genes that were thought to only be switched on in the cells of specific tissues are also on in the endothelial cells lining the blood vessels of the tissue. For example, genes switched on in brain cells are also active in brain endothelial cells, and genes allowing heart muscle cells to pump are also on in the endothelial cells of the heart blood vessels.

The endothelial cell genetic signatures identified by Jambusaria et al. can be used as "postal codes" to target drugs to a specific organ via the endothelial cells that feed it. It might also be possible to use these genetic signatures to build organ-specific blood vessels from stem cells in the laboratory. Future work will try to answer why endothelial cells serving the heart and brain use genes from these organs.

hand, are specialized in a manner to ensure ready supply of fatty acids to voracious cardiomyocytes which rely on continuous supply of fatty acids as the primary fuel to generate ATP necessary for cardiac contraction (*Potente and Mäkinen, 2017*).

Identifying differences in the expression levels of selected genes in endothelial cells from different tissues or organs provides some insights into the molecular underpinnings of endothelial heterogeneity, however unbiased gene expression profiling is likely to yield a more comprehensive evaluation of the genes and regulatory pathways underlying endothelial heterogeneity. Microarray profiling has been used to identify paracrine factors and signaling pathways that characterize endothelial cells in different organs (*Jambusaria et al., 2018*; *Nolan et al., 2013*). Single-cell transcriptomic analysis of endothelial cells has also provided a molecular atlas of the brain and lung vasculature at a single cell level (*Vanlandewijck et al., 2018*). The latter work has characterized transcriptomic signatures of distinct endothelial subpopulations. While single cell RNA-sequencing is ideally suited for identifying subpopulations within a single vascular bed, current single cell technologies are limited in their ability to detect the expression of individual genes in a given cell (*Bacher and Kendziorski, 2016*; *Zhu et al., 2018*; *Kharchenko et al., 2014*; *Lun et al., 2016*; *Vallejos et al., 2017*). The endothelial signatures defined using these transcriptomic approaches are potentially influenced by disassociation and isolation of endothelial cells, a process affecting cellular mRNA levels when cells are removed from their native niche (*Haimon et al., 2018*; *Rossner et al., 2006*; *Sugino et al., 2006*). Furthermore, conventional global mRNA and single cell mRNA transcriptomic profiling does not discriminate between the total mRNA pool and those mRNAs preferentially translated due to translational regulation (*Zhou et al., 2016*; *Piccirillo et al., 2014*).

In the present study, to understand further the variegated nature of the endothelium, we used the RiboTag transgenic mouse model, in which LoxP mice express an HA-tag on the ribosomal Rpl22 protein (*Sanz et al., 2009*). These mice enable direct isolation of tissue-specific mRNAs

undergoing translation without cell disassociation (*Sanz et al., 2009*). Using an endothelial-specific RiboTag model, we show that organ-specific ECs have distinct translatome patterns of gene clusters during homeostasis. Since the circulating bacterial endotoxin lipopolysaccharide (LPS) is a key mediator of tissue inflammation and injury in patients with bacteremia and sepsis (*Cross, 2016*) (*Charbonney et al., 2016*), we also exposed the RiboTag mice to LPS to induce systemic inflammatory injury and studied the organ-specific EC translatome response. We found that ECs express tissue-specific genes involved in vascular barrier function, metabolism, and substrate-specific transport. In addition, we found that ECs expressed genes thought to be primarily expressed in the surrounding tissue parenchyma, suggesting a previously unrecognized organ-specific endothelial plasticity and adaptation. To allow other researchers to explore the organ-specific EC translatome heterogeneity, we have generated a searchable database (http://www.rehmanlab.org/ribo), in which users can visualize gene expression levels of individual genes.

## Results

### Optimized platform to characterize organotypic endothelial heterogeneity

To precisely investigate the in-situ organ-specific EC molecular signature in brain, lung, and heart tissue we crossed the RiboTag mice (Rpl22$^{HA/+}$) (*Sanz et al., 2009*) with the endothelial-specific VE-cadherin-Cre mice (*Jeong et al., 2017*; *Sörensen et al., 2009*) to generate RiboTag$^{EC}$ (*Cdh5$^{CreERT2/+}$; Rpl22$^{HA/+}$*) mice. At 4 weeks post tamoxifen administration, ribosomes in the endothelial cells of all tissues expressed the HA tag, thus allowing for the specific isolation of mRNA undergoing ribosomal translation from ECs in the brain, heart and lung during homeostatic conditions. We also isolated brain, lung, and heart endothelial mRNA at several time points following systemic inflammatory injury, induced using a sublethal dose of the bacterial endotoxin lipopolysaccharide (LPS), ranging from the acute injury phase at 6 hr post-LPS to the recovery phase at 1 week post-LPS (*Figure 1—figure supplement 1A*). Log fold change (logFC) values were calculated between endothelial mRNA (immunoprecipitated by an anti-HA antibody) versus whole tissue mRNA (immunoprecipitated with control antibody, anti-RPL22) using quantitative PCR (qPCR). The analysis of the qPCR data confirmed enrichment of endothelial-specific RNA similar to what has been reported in other studies using the RiboTag model (*Jeong et al., 2017*) and also demonstrated minimal expression of RNA from other tissue-resident cell types (*Figure 1—figure supplement 1B–1F*).

After confirming the enrichment of endothelial RNA using qPCR, we performed global transcriptional profiling with RNA-Seq on the RiboTag$^{EC}$ brain, lung, and heart samples. Principal component analysis (PCA) of the RNA-Seq data for endothelial mRNA from brain, lung, and heart tissue from all time points showed a clear separation between the replicate brain, lung, and heart translatomes, indicating that ECs from each tissue demonstrated a distinct transcriptional identity at baseline that is maintained even in the setting of profound systemic inflammatory injury (*Figure 1A*). In order to identify the genes responsible for these distinct tissue-specific EC profiles, we performed a differential expression analysis on the RNA-Seq data. The differential expression analysis was concordant with the PCA and identified 1692 genes which were differentially expressed in brain ECs (versus ECs from the other two tissues), 1052 genes which were differentially expressed in lung ECs, and 570 genes which were differentially expressed in heart ECs (*Figure 1B*).

We next analyzed the baseline heterogeneity of ECs obtained from brain, lung and heart by assessing the gene expression levels of endothelial genes using established databases. We specifically focused our analysis on a pan-endothelial gene set (*Franzén et al., 2019*), glycolysis and fatty acid metabolism gene sets (*Shimoyama et al., 2015*) and a solute transport gene set (*Hediger et al., 2013*). Hierarchical clustering of the RNA-Seq profiles on merely 152 pan-endothelial genes from PanglaoDB (*Franzén et al., 2019*) separated all replicate baseline samples, indicating that classical endothelial markers are sufficient to differentiate ECs from these three organs (*Figure 1C*). For example, genes upregulated in brain ECs included T-box transcription factor (*Tbx1*) and the glucose transporter 1 (*Slc2a1*), genes upregulated in the lung endothelium included claudin 5 (*Cldn5*) and the Hes related family BHLH transcription factor with YRPW Motif 1 (*Hey1*), whereas

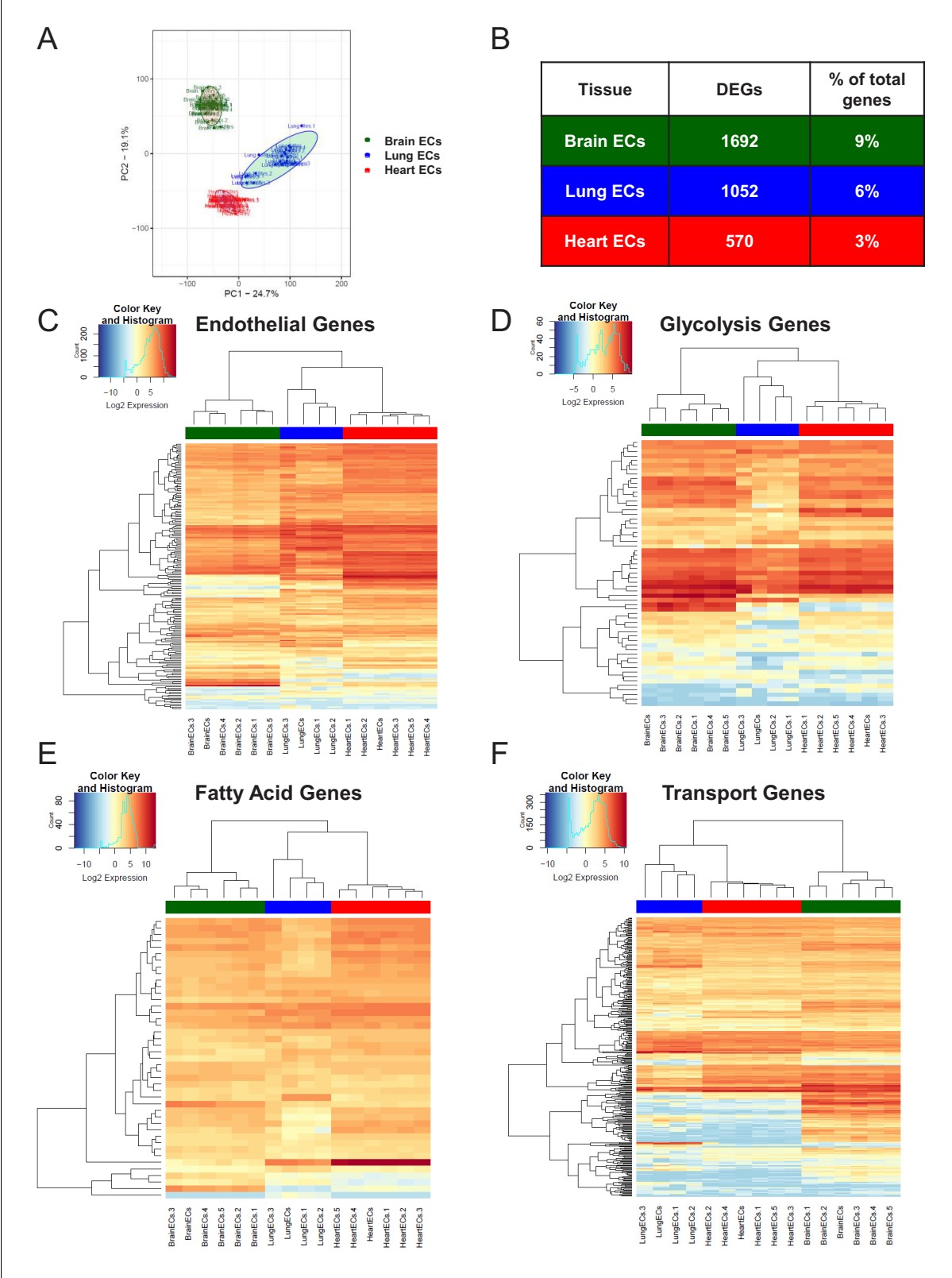

**Figure 1.** Endothelial heterogeneity exists in classic endothelial functions. (**A**) Principal component analysis of RNA-Seq data generated from brain, lung, and heart endothelial samples isolated from RiboTag[EC] mice displays the organ-specific in-situ endothelial clusters. (**B**) Differential expression analysis of 18,910 genes which are expressed in brain, lung, and heart endothelium at baseline identified tissue-specific differentially expressed genes. *Figure 1 continued on next page*

Figure 1 continued

(FDR < 5%) (C–F) Hierarchical clustering of classical endothelial processes including (C) endothelial genes, (D–E) metabolism, and (F) transporters results in distinct clustering of brain, lung, and heart endothelial baseline samples.

The online version of this article includes the following figure supplement(s) for figure 1:

**Figure supplement 1.** RiboTag Isolation of endothelial mRNA.
**Figure supplement 2.** Characterization of Whole Brain RNA-Seq data.
**Figure supplement 3.** Characterization of Whole Lung RNA-Seq data.
**Figure supplement 4.** Characterization of Whole Heart RNA-Seq data.
**Figure supplement 5.** Kendall's Tau correlation supports endothelial mRNA isolation from RiboTag[EC] mice.

heart ECs demonstrated upregulation of vascular endothelial growth factor receptor 2 (*Kdr*) and the endothelial cell surface expressed chemotaxis and apoptosis regulator (*Ecsr*).

We next focused on the tissue-specific upregulation of metabolic genes. As seen in the glycolysis gene heatmap, we found that most tissue-specific EC genes involved in glycolysis were specifically upregulated in the brain endothelium (*Figure 1D*), but there were selected glycolytic genes specifically upregulated in other tissues such as 6-phosphofructo-2-kinase/fructose-2,6-biphosphatase 3 (*Pfkfb3*) in lung ECs and alcohol dehydrogenase 1 (*Adh1*) in heart ECs. In contrast, fatty acid metabolism genes were most upregulated in heart ECs consistent with the heavy reliance of the heart on fatty acids to generate ATP (*Figure 1E*). Heart ECs exhibited upregulation of 17 fatty acid metabolism genes whereas brain ECs and lung ECs only demonstrated upregulation of 9 and 4 metabolism genes, respectively.

Regarding solute transport genes, the brain endothelium showed the most specific upregulation of genes when compared to ECs of the other tissues, both in terms of number of transporters as well as the magnitude of upregulation. We found that 141 transporter genes were upregulated in brain ECs, whereas 43 and 44 genes were upregulated in lung and heart ECs, respectively. As seen in the heatmap (*Figure 1F*), the expression levels of brain EC-specific transporters were far greater than those of lung and heart ECs, indicative of the central role of solute transport regulation in brain EC function.

## RiboTag[EC] endothelial mRNA purity

After confirming the efficiency of the RiboTag immunoprecipitation protocol using qPCR, we next sought to perform an unbiased and systematic analysis of the utility of the RiboTag[EC] model as a tool to study the organ-specific endothelial translatome heterogeneity. We therefore compared organ-specific RiboTag[EC] RNA-Seq baseline profiles to healthy whole-tissue RNA-Seq profiles obtained from publicly available whole tissue RNA-Seq datasets (*Athar et al., 2019*). By applying normalization and batch correction techniques, we were able to directly compare the mRNA expression levels of RiboTag[EC] endothelial samples with those of whole tissue samples.

To characterize the whole brain, lung, and heart samples, we identified the genes that were significantly upregulated in each of the tissues and generated a heatmap displaying the 1358 differentially upregulated whole brain-specific genes relative to whole lung and whole heart (*Figure 1—figure supplement 2A*). By performing a gene set enrichment analysis (GSEA) to ascertain the pathways associated with these genes, we confirmed the validity of the samples because the top pathways included 'neurotransmitter transport', 'synapse organization', 'synaptic vesicle cycle' (*Figure 1—figure supplement 2B*). The top 10 most abundant genes in the whole brain RNA-Seq data included myelin basic protein (*Mbp*), proteolipid protein 1 (*Plp1*), calmodulin 1 (*Calm1*), synaptosome associate protein 25 (*Snap25*), kinesis family member 5A (*Kif5a*), ATPase Na+/K+ transporting subunit alpha 3 (*Atp1a3*), sodium-dependent glutamate/aspartate transporter 2 (*Slc1a2*), secreted protein acidic and cysteine rich (*Sparcl1*), carboxypeptidase e (*Cpe*), stearoyl-coA desaturase 2 (*Scd2*) (*Figure 1—figure supplement 2C*).

Whole lung samples were characterized by 1071 differentially expressed genes (*Figure 1—figure supplement 3A*) on which we performed GSEA (*Figure 1—figure supplement 3B*). The top 10 most abundant genes in the whole lung were desmoyokin (*Ahnak*), microtubule-actin crosslinking factor 1 (*Macf1*), actin beta (*Actb*), surfactant protein c (*Sftpc*), spectrin beta, non-erythrocytic 1 (*Sptbn1*), hypoxia inducible factor two alpha (*Hif2a*), stearoyl-CoA desaturase (*Scd1*), filamin a (*Flna*), adhesion

g protein-coupled receptor f5 (*Adgrf5*), and ldl receptor related protein 1 (*Lrp1*) (*Figure 1—figure supplement 3C*).

The signature of the whole heart derived from differential gene expression analysis was composed of 1351 genes (*Figure 1—figure supplement 4A*). GSEA indicated a preponderance of metabolic and muscle contraction pathways (*Figure 1—figure supplement 4B*). The top 10 most abundant cardiac genes were myosin heavy chain 6 (*Myh6*), ATPase sarcoplasmic/endoplasmic reticulum $Ca^{2+}$ transporting 2 (*Atp2a2*), myoglobin (*Mb*), actin, alpha, cardiac muscle 1 (*Actc1*), phospholamban (*Pln*), myosin regulatory light chain 2 (*Myl2*), titin (*Ttn*), troponin t2, cardiac type (*Tnnt2*), tropomyosin 1 (*Tpm1*), and lipoprotein lipase (*Lpl*) (*Figure 1—figure supplement 4C*).

After establishing and confirming the molecular signatures of the whole brain, whole lung, and whole heart tissue, we next calculated a Kendall's Tau correlation coefficient to assess the rank correlation between the RiboTag^EC samples and the whole tissue samples. We surmised that if the rank of the most abundant whole tissue genes was the same as the rank of these genes in the RiboTag^EC samples, then it would indicate possible contamination of the EC samples with whole tissue mRNA; however, if the abundance rank order of whole tissue genes was quite distinct from that in the RiboTag^EC samples, then it would indicate tissue specific programming of ECs in situ (*Figure 1—figure supplement 5A*). We assessed the Kendall's Tau rank correlation for all three tissues and plotted correlation heatmaps showing the results (*Figure 1—figure supplement 5B–D*). Our findings indicate that there was no significant correlation between the abundance rank of whole tissue genes and their rank order in the RiboTag^EC samples. The rank correlation in the brain samples ranged from −0.29 to 0.38 (*Figure 1—figure supplement 5B*). Since the cellular composition of the lung is 40–50% endothelial, we expectantly saw a higher rank correlation between whole lung samples and lung RiboTag^EC samples, ranging between 0.02 and 0.6 (*Figure 1—figure supplement 5C*). In the heart, we found a range of rank correlations between −0.29 to 0.24 (*Figure 1—figure supplement 5D*). These results provide mathematical evidence for the robustness and purity of the RiboTag^EC samples.

## Brain-specific endothelial molecular signature

After confirming the robustness and purity of the RiboTag^EC samples, we performed differential expression analysis to identify the significantly upregulated genes in the brain endothelial translatome (*Figure 2A*, *Supplementary file 1*). We used these upregulated genes as the input into GSEA to characterize the brain ECs (*Figure 2B*). Surprisingly, we found that genes involved in processes typically thought of being canonical neuronal functions such as synapse organization, neurotransmitter transport, axon development, and regulation of ion transmembrane transport were significantly enriched in brain ECs (*Figure 2B*). The top 10 most significantly upregulated genes in the brain ECs included: prostaglandin d synthase (*Ptgds*), ATPase, Na+/K+ transporting, alpha two polypeptide (*Atp1a2*), basigin (*Bsg*), apolipoprotein e (*Apoe*), glutamate-ammonia ligase (*Glul*), apolipoprotein d (*Apod*), pleiotrophin (*Ptn*), insulin like growth factor 2 (*Igf2*), osteonectin (*Spock2*), and glucose transporter 1 (*Slc2a1*) (*Figure 2C*). In order to identify brain EC-specific surface markers, which could be of great value for therapeutic targeting of brain ECs, we used the Cell Surface Protein Atlas database (*Bausch-Fluck et al., 2015*) and identified the top 10 surface markers for brain ECs (*Figure 2D*), which included the glutamate/aspartate transporter II (*Slc1a2*), thyroxine transporter (*Slco1c1*), glial fibrillary acidic protein (*Gfap*), ATPase Na+/K+ transporting subunit alpha 3 (*Atp1a3*), endothelin b receptor-like protein 2 (*Gpr37l1*), Delta/Notch like EGF repeat containing transmembrane (*Dner*), synaptic vesicle glycoprotein 2b (*Sv2b*), sodium voltage-gated channel beta subunit 2 (*Scn2b*), glutamate ionotropic receptor NMDA type subunit 2a (*Grin2a*), and neurofascin (*Nfasc*). Individual boxplots for the $\log_2$ expression levels of each gene show that the expression levels of these cell surface markers are 6–8 $\log_2$ units higher in brain ECs than in the lung and heart endothelium. We freshly isolated individual ECs, performed a cytospin and stained for the neurotrophic factor PTN and found that it was expressed on individual brain ECs but at much lower levels in heart or lung ECs (*Figure 2E*).

## Lung-specific endothelial molecular signature

We next analyzed the lung EC signature using differential expression analysis (*Figure 3A*). We found that the lung endothelium exhibits significant upregulation of genes involved in biological processes

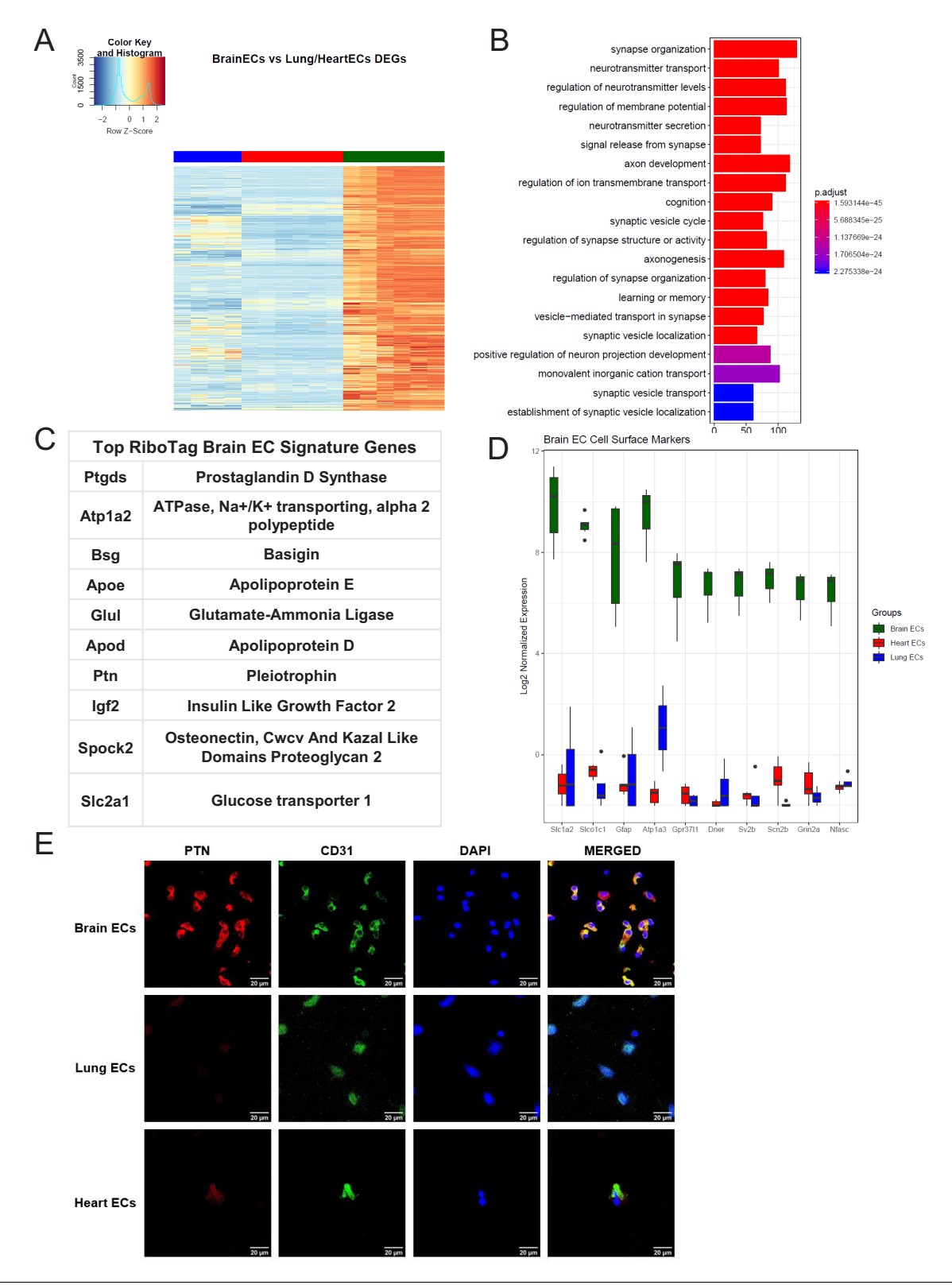

**Figure 2.** Brain endothelial specific signature. (**A**) Heat map representation of differentially upregulated genes identified by comparing brain ECs to lung and heart ECs at baseline. The blue to white to red gradient represents increasing expression of the pathway with blue representing minimal expression while the red represents high expression of the pathway. Individual gene expression values can be visualized at www.rehmanlab.org/ribo (**B**) The GSEA results of enriched GO terms from RiboTag brain ECs at baseline. (**C**) Top RiboTag brain EC signature markers ranked in order of logFC. (**D**) *Figure 2 continued on next page*

*Figure 2 continued*

Top RiboTag brain EC cell surface markers identified using the Cell Surface Protein Atlas. (E) Confocal analysis was performed after brain, lung, and heart ECs were processed on a cytospin to assess brain EC PTN (Pleotrophin) specificity. A scale bar of 20 μm is included on all images.

related to immune function such as leukocyte cell-cell adhesion, T cell activation, leukocyte migration, and regulation of immune system processes (*Figure 3B*). The 10 most significantly upregulated genes in lung ECs included surfactant protein c (*Sftpc*), advanced glycosylation end-product specific receptor (*Ager*), norepinephrine transporter (*Slc6a2*), chitinase-like protein 3 (*Chil3*), WAP four-disulfide cco domain 2 (*Wfdc2*), c-type lectin domain containing 7a (*Clec7a*), mucin 1 (*Muc1*), resistin like alpha (*Retnla*), lysozyme (*Lyz1*), homeobox a5 (*Hoxa5*) (*Figure 3C*). The top lung endothelial cell surface markers included norepinephrine transporter (*Slc6a1*), mucin 1 (*Muc1*), tumor necrosis factor c (*Ltb*), prostaglandin transporter (*Slco2a1*), epithelial membrane protein 2 (*Emp2*), ATPase sarcoplasmic/endoplasmic reticulum Ca2+ transporting 3 (*Atp2a3*), epithelial cell adhesion molecule (*Epcam*), leukocyte function-associated molecule one alpha chain (*Itgal*), interleukin three receptor subunit alpha (*Il3ra*), matriptase (*St14*) (*Figure 3D*). We validated our computational analysis by staining freshly isolated ECs for RAGE and found that RAGE was only expressed at significant levels in lung ECs but not heart or brain ECs (*Figure 3E*).

## Heart-specific endothelial molecular signature

We then studied the differentially expressed genes in the heart endothelium (*Figure 4A*, *Supplementary file 3*). GSEA identified pathways specifically upregulated in heart ECs, as compared to brain and lung ECs (*Figure 4B*). Strikingly, we found that the genes specifically upregulated in heart ECs were involved in processes such as cardiac muscle tissue development, myofibril assembly and cardiac contraction (*Figure 4B*). This suggested that the cardiac endothelium expresses genes canonically thought to be cardiomyocyte genes, analogous to the expression of canonical neuronal genes in the brain endothelium. The top expressing heart EC genes included myosin regulatory light chain 2 (*Myl2*), myosin regulatory light chain 3 (*Myl3*), aquaporin 7 (*Aqp7*), ADP-ribosylhydrolase like 1 (*Adprhl1*), alpha 2-HS glycoprotein (*Ahsg*), sodium-coupled nucleoside transporter (*Slc28a2*), xin actin binding repeat containing 2 (*Xirp2*), myoglobin (*Mb*), Butyrophilin like 9 (*Btnl9*), creatine kinase, mitochondrial 2 (*Ckmt2*), leucine rich repeats and transmembrane domains 1 (*Lrtm1*), and fatty acid binding protein 4 (*Fabp4*) (*Figure 4C*).The top 10 heart EC surface marker genes included alpha 2-HS glycoprotein (*Ahsg*), sodium-coupled nucleoside transporter (*Slc28a2*), titin (*Ttn*), tumor necrosis factor receptor superfamily member 27 (*Eda2r*), platelet glycoprotein 4 (*Cd36*), laminin subunit alpha 4 (*Lama4*), fibulin 2 (*Fbln2*), ectonucleotide pyrophosphatase/phosphodiesterase 3 (*Enpp3*), t-cadherin (*Cdh13*), steroid sensitive gene 1 (*Ccdc80*) (*Figure 4D*). We tested the heart EC specificity of AQP7 using confocal analysis on freshly isolated brain, lung, and heart ECs and found that AQP7 was robustly expressed in heart ECs but minimally expressed in brain and lung ECs (*Figure 4E*).

## Single-cell endothelial heterogeneity

In light of the surprising findings that endothelial cells express genes typically associated with surrounding parenchymal cells such as cardiomyocytes or neuronal cells, we next used single cell RNA-Seq analysis to assess whether the RiboTag^EC endothelial signatures are also found in individual endothelial cells by analyzing endothelial single-cell data from the Tabula Muris compendium (*Tabula Muris Consortium et al., 2018*) and the single cell RNA-Seq atlas of the brain and lung endothelium (*Vanlandewijck et al., 2018*). Using expression of the endothelial genes *Cd31* and *Cdh5* as markers of ECs, we analyzed double positive cells for both markers in Tabula Muris brain, lung, and heart tissues and performed PCA to assess the extent of endothelial heterogeneity (*Figure 5A*). The PCA plot partitioned the cells into groups defined by their tissue of origin, indicating a tissue-specific EC signature even at the single cell level. Similarly, we performed PCA on ECs in Betsholtz dataset (which relied on *Cd31* and *Cldn5* as EC markers) and also found that ECs similarly clustered according to their tissue of origin (*Figure 5B*).

We then used these two scRNA-Seq endothelial datasets for the three organs we had analyzed in our RiboTag experiments and intersected the differentially expressed genes for each organ-specific endothelial population. The intent of this was to ascertain which tissue-specific EC signature genes

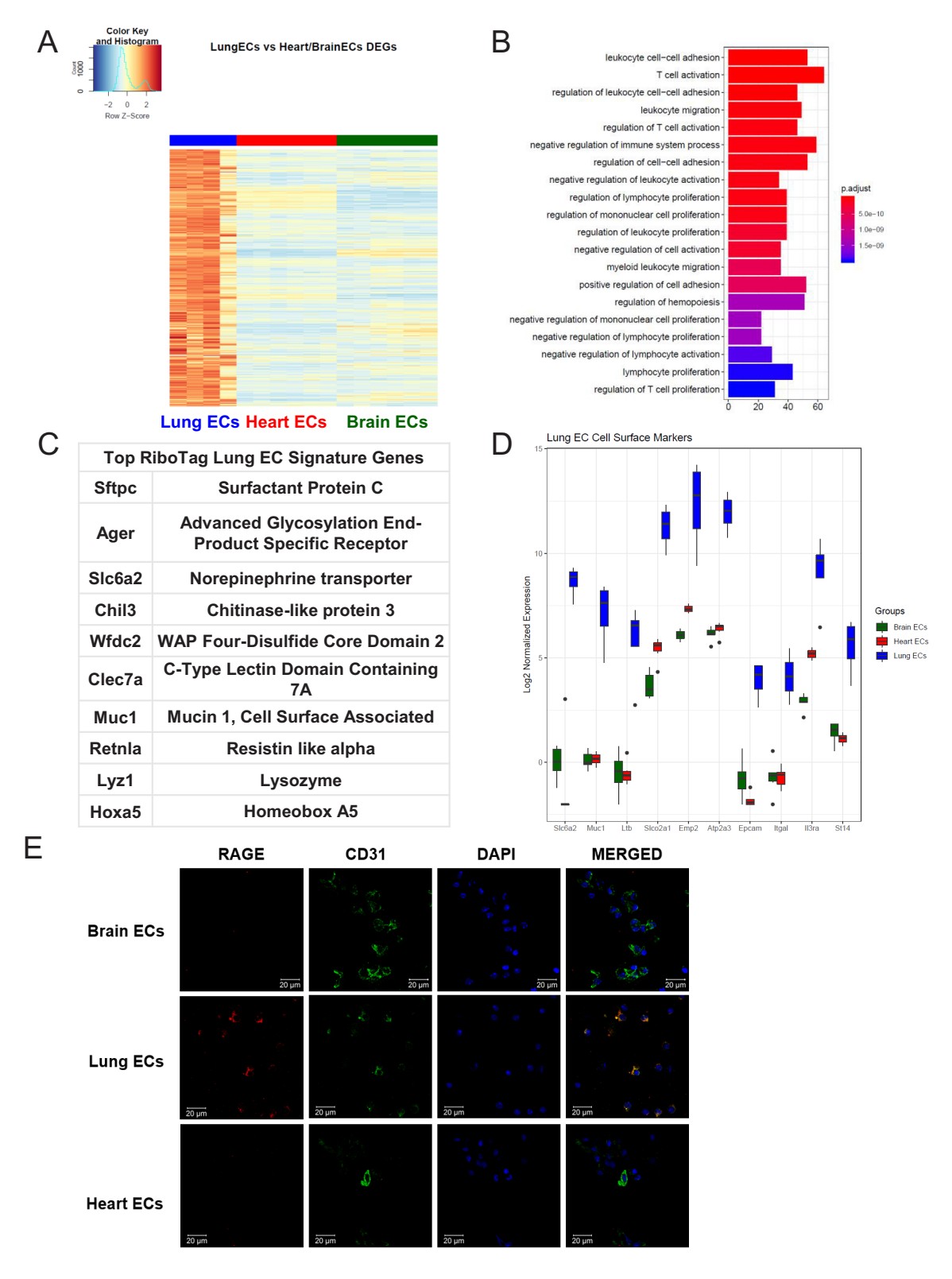

**Figure 3.** Lung endothelial specific signature. (**A**) Heat map representation of differentially upregulated genes identified by comparing lung ECs to brain and heart ECs at baseline. The blue to white to red gradient represents increasing expression of the pathway with blue representing minimal expression while the red represents high expression of the pathway. Individual gene expression values can be visualized at www.rehmanlab.org/ribo (**B**) The GSEA results of enriched GO terms from RiboTag lung ECs at baseline. (**C**) Top RiboTag lung EC signature markers ranked in order of logFC. (**D**) *Figure 3 continued on next page*

*Figure 3 continued*

Top RiboTag lung EC cell surface markers identified using the Cell Surface Protein Atlas. (E) Confocal analysis was performed after brain, lung, and heart ECs were processed on a cytospin to assess lung EC RAGE (Receptor for Advanced Glycation Endproducts) specificity. A scale bar of 20 μm is included on all images.

were present in the single cell datasets as well as our RiboTag[EC] dataset. We found that the shared brain EC signature across all three datasets (Tabula Muris[EC], Betsholtz[EC] and RiboTag[EC]) for brain ECs was enriched for genes involved in ion transport, acid transport, synapse organization and neurotransmitter transport (*Figure 5C*). This finding is consistent with the brain EC-specific enrichment of neuronal signaling pathways that had been identified by the RiboTag[EC] analysis (*Figure 2*). We also found that the genes specifically upregulated in the Tabula Muris and Betsholtz lung ECs were involved in T cell activation, TGFβ signaling, and antigen processing and presentation (*Figure 5D*), again consistent with the 'immune activation' signature identified by the RiboTag[EC] analysis alone (*Figure 3*). Similarly, the shared upregulated genes in Tabula Muris single cell heart ECs were involved in processes such as cardiac muscle contraction, myofibril assembly and proliferation (*Figure 5E*, *Figure 4*).

We next quantified the intersection of brain, lung and heart endothelial marker genes across the Tabula Muris, brain and lung EC atlas, and RiboTag datasets. For the brain endothelium, 40 of the Tabula Muris top 50 brain EC specific genes were also brain EC specific genes in the RiboTag dataset. In the Betsholtz dataset, 27 of the top 50 brain EC specific genes were present in the RiboTag brain EC specific genes (*Figure 5F*). We found that 17 of the top lung endothelial specific genes in the Betsholtz data set were also found in the list of lung endothelial-specific genes in the RiboTag model (*Figure 5G*). Of the 24 top lung endothelial specific genes found in the Tabula Muris data set, the same genes were also found in the list of lung endothelial-specific genes in the RiboTag model (*Figure 5G*).

## Organ-specific parenchymal gene signature exists in endothelial scRNA-Seq

To address further that the parenchymal signatures (*Supplementary files 4–6*) identified in the endothelial translatome were simply not driven by low abundance of transcripts, we performed a Spearman correlation analysis to compare organ-matched RiboTag bulk RNA-Seq data with scRNA-Seq data generated by the Betsholtz and the Tabula Muris Compendium (*Figure 6*, *Figure 6—figure supplement 1*). In each dataset, we first determined the fold change for all genes using a housekeeping gene, *Sap30l* which we identified as being stably expressed across all datasets, and thus ideally suited to perform relative abundance comparisons (*Supplementary files 7–9*). Using the fold change values, we calculated the correlation coefficients between the brain endothelial translatome and single cell brain ECs from the Betsholtz and Tabula Muris datasets. We found that the correlation between RiboTag and Betsholtz was 0.53 for all genes detected in the brain endothelium (*Figure 6A*) while the correlation between RiboTag and Tabula Muris was 0.47 (*Figure 6—figure supplement 1A*). We then specifically tested whether the parenchymal signature genes in the brain endothelium were correlated with the Betsholtz and Tabula Muris individual brain ECs. The correlation of the parenchymal gene expression between RiboTag brain EC samples and Betsholtz brain ECs was 0.31 (*Figure 6B*) while with Tabula Muris brain ECs the correlation was 0.28 (*Figure 6—figure supplement 1B*). Importantly, the brain EC parenchymal genes including synaptosome associated protein 47 (*Snap47*) and synaptotagmin 11 (*Syt11*) were expressed at similar or higher levels in the single cell brain ECs from the Betsholtz and Tabula Muris datasets than in the RiboTag brain EC samples (*Figure 6C*). We performed identical analysis for the lung and heart endothelium (*Figure 6D–I*, *Figure 6—figure supplement 1*), and found that similar correlation values ranging between 0.37 to 0.68. Of note, the heart endothelial gene expression was the most correlated organ across the distinct platforms (*Figure 6G–H*). In the lung and heart endothelium, we also found that individual genes representing the parenchymal signature were expressed at similar or higher levels in the single cell samples (*Figure 6F*, *Figure 6G–I*), such as the cardiac contractile gene Tropomyosin (*Tpm1*), which was expressed at higher levels in individual heart ECs from the Tabula Muris dataset.

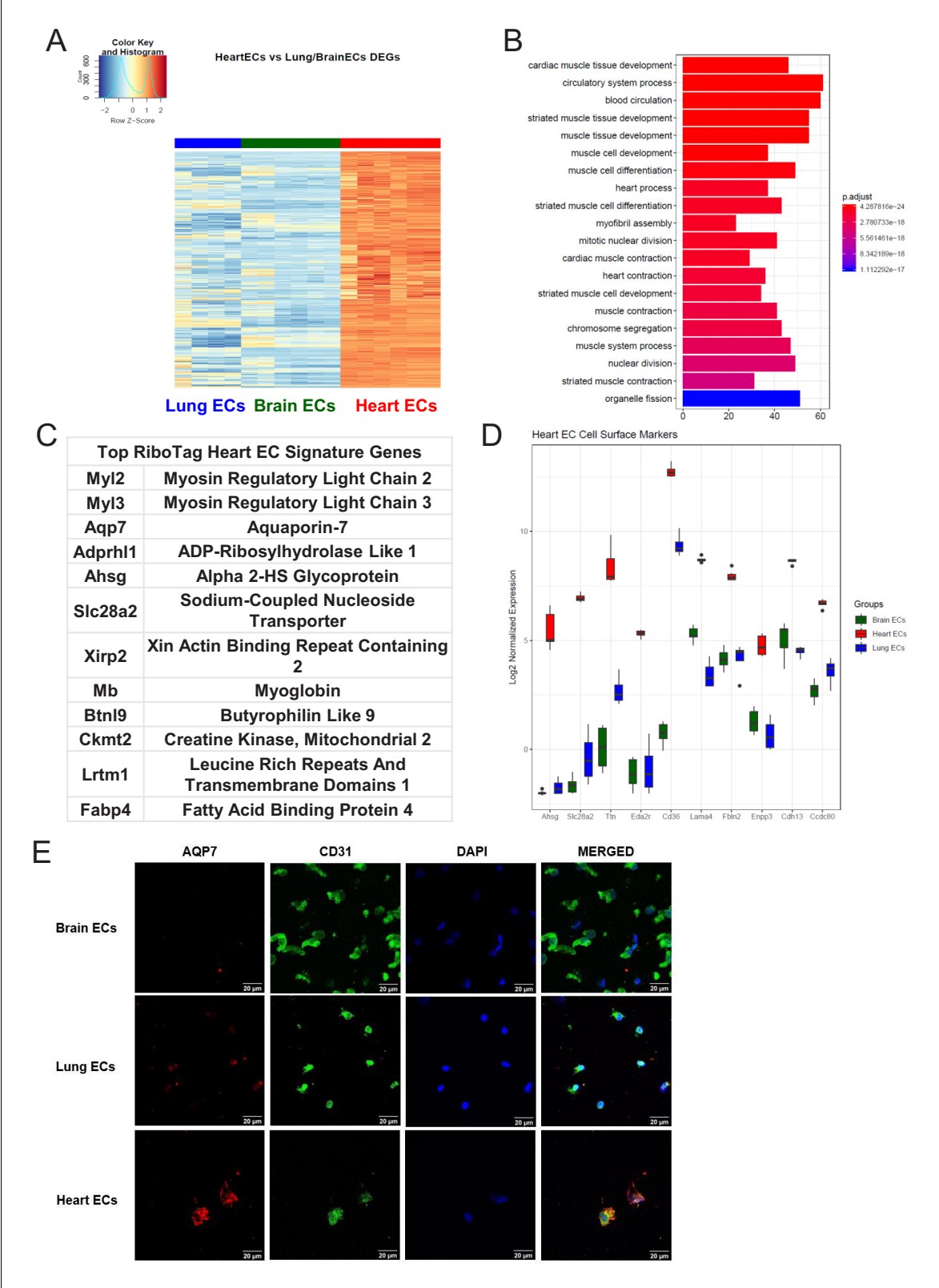

**Figure 4.** Heart endothelial specific signature (**A**) Heat map representation of differentially upregulated genes identified by comparing heart ECs to brain and lung ECs at baseline. The blue to white to red gradient represents increasing expression of the pathway with blue representing minimal expression while the red represents high expression of the pathway. Individual gene expression values can be visualized at www.rehmanlab.org/ribo (**B**) The GSEA results of enriched GO terms from RiboTag heart ECs at baseline. (**C**) Top RiboTag heart EC signature markers ranked in order of logFC. (**D**) *Figure 4 continued on next page*

*Figure 4 continued*

Top RiboTag heart EC cell surface markers identified using the Cell Surface Protein Atlas. (E) Confocal analysis was performed after brain, lung, and heart ECs were processed on a cytospin to assess heart EC AQP7 (Aquaporin 7) specificity. A scale bar of 20 µm is included on all images.

## In situ organ-specific endothelial early- and late-inflammation signature

We next analyzed the dynamics of the EC inflammatory response in each tissue, focusing on the early response (6 hr post systemic LPS) and late response (24 hr post systemic LPS). At these time points, we identified the genes most upregulated by inflammatory injury in each tissue (*Figure 8—figure supplement 1*). In the brain endothelium, we identified several differentially expressed acute inflammatory factors including selectins, chemokine receptors, and interleukins which were strongly activated 6 hr post LPS treatment (*Figure 7A–C*). We analyzed the kinetics during the entire time course for the early inflammatory brain endothelial specific genes such as eosinophil chemotactic protein (*Ccl11*) (*Figure 7C*) and found that *Ccl11* is markedly upregulated at the 6 hr time point and remains significantly higher in the brain endothelium, but by one week post LPS injection the expression level returns to the same level as that seen in lung and heart endothelium. In the lung endothelium, we discovered that the most upregulated inflammatory pathways included chemokines, response to cellular stress, hematopoiesis genes and early immune response mediators (*Figure 7D–F*). Lymphocyte antigen 96 (*Ly96*) was strongly upregulated (*Figure 7D*) whereas the apoptosis gene caspase 6 (*Casp6*) was markedly downregulated 6 hr post LPS treatment and remained lower in lung ECs than in brain or heart ECs throughout the injury period (*Figure 7F*). In heart ECs, leukocyte migration and neutrophil activation pathways were most upregulated by inflammatory injury (*Figure 7G–I*). At 24 hr post injury, we found the peak upregulation of inflammatory genes (*Figure 8*) with a substantial overlap of the inflammatory response pathways, predominantly associated with neutrophil and leukocyte chemotaxis and migration, in the brain (*Figure 8A–C*), lung (*Figure 8D–F*), and heart ECs (*Figure 8G–I*).

## Tissue-specific dynamic response following LPS-induced inflammatory activation

After establishing the baseline heterogeneity of brain, lung and heart ECs, we next studied the dynamics of the organ-specific baseline endothelial signature during systemic inflammation, we collected translatome data of the brain, lung, and heart endothelium at several time points following LPS treatment. By computationally analyzing RiboTag$^{EC}$ mRNA from brain, lung, and heart at 0 hr, 6 hr, 24 hr, 48 hr, 72 hr, and 168 hr post-LPS administration, we were able to identify tissue-specific molecular mechanisms modulated in endothelial injury, repair, and regeneration.

We first investigated the tissue-specific baseline signatures over time in order to address the question of whether the baseline core endothelial functions were disrupted during inflammatory activation. The time-course of the brain endothelium specific endothelial genes were plotted to compare their kinetics to the lung and heart endothelium (*Figure 9A*). We found that selected genes which constitute the tissue-specific EC signature during homeostasis are modulated during inflammatory injury. For instance, the expression level of von Willebrand factor A domain containing protein 1 (*Vwa1*) which we found to be a brain endothelial gene during homeostasis decreases during early and late inflammation and then returns to baseline levels one-week post LPS injury, whereas its levels in lung and heart endothelium remain relatively low during the entire time course. On the other hand, there are signature genes such as glucose transporter protein 1 (*Slc2a1*) which is consistently upregulated in brain ECs throughout the post-injury period.

From the analysis of the lung endothelium specific endothelial genes heatmap (*Figure 9B*), it is apparent that expression of nearly all the canonical endothelial genes drastically decrease during the early and late inflammatory time points. This is an important finding because it suggests that the lung endothelium experiences the most profound dysregulation of core endothelial genes following LPS injury. We also identified lung endothelial specific genes which are solely modulated in the lung endothelium during the inflammatory time course. For instance, the expression levels of forkhead-related transcription factor 1 (*Foxf1*) and tetraspanin8 (*Tspan8*) significantly decrease in the lung endothelium at 6 hr and 24 hr post LPS treatment and then gradually recover back to baseline levels, but both genes remain minimally expressed in the brain and heart endothelium.

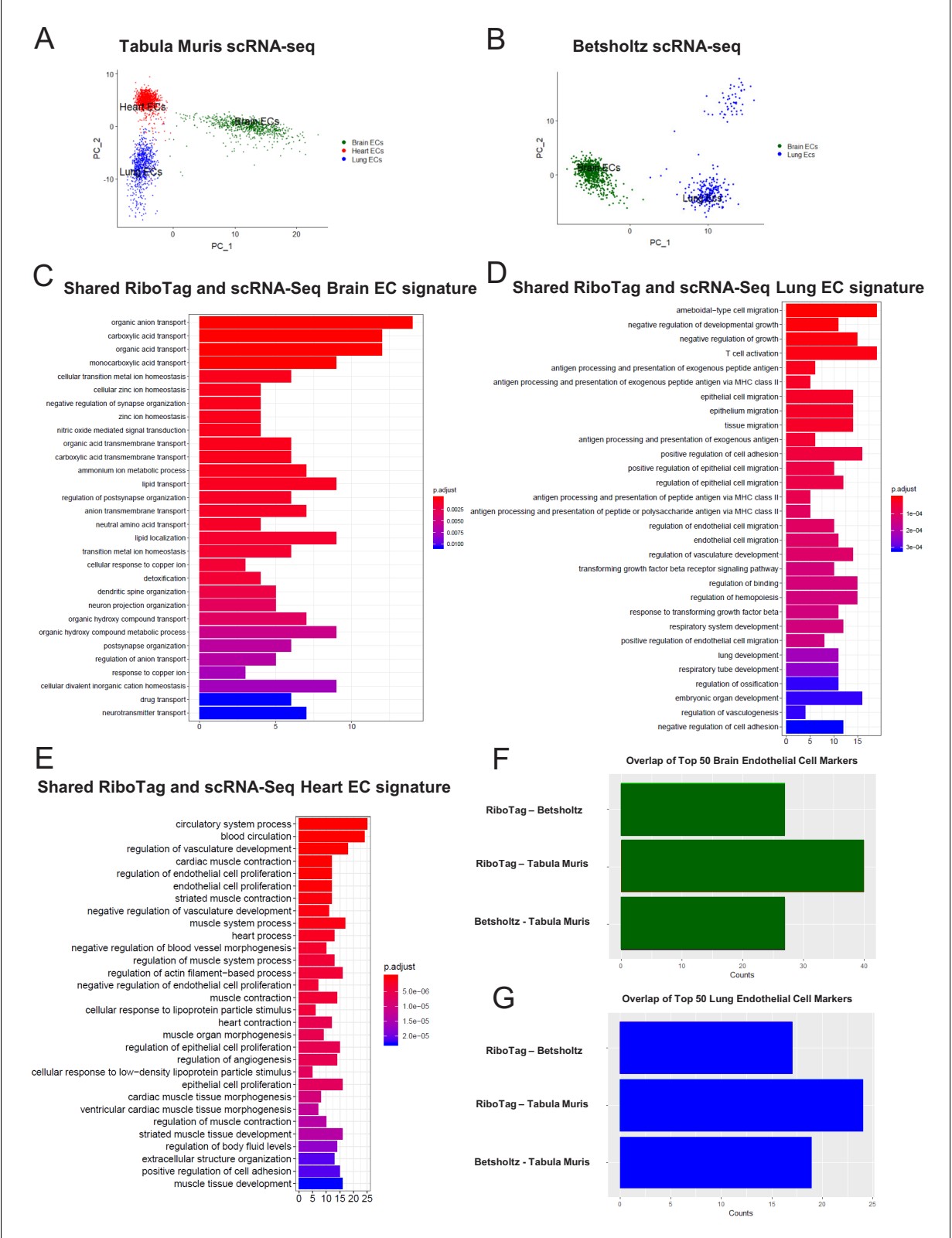

**Figure 5.** Single-cell endothelial heterogeneity (A) PCA of endothelial scRNA-Seq data from the Tabula Muris collection of mouse tissues colored by tissue. (B) PCA of endothelial scRNA-Seq data from the Betsholtz Lab of mouse tissues colored by tissue. The GSEA results of enriched GO terms from overlapping differentially expressed genes between RiboTag and Betsholtz or Tabula Muris for (C) brain ECs, (D) lung ECs, and (E) heart ECs. (F)
*Figure 5 continued on next page*

Figure 5 continued

Overlap of top 50 scRNA-Seq brain EC marker genes with RiboTag brain EC marker genes. (**G**) Overlap of top 50 scRNA-Seq lung EC marker genes with RiboTag lung EC marker genes.

The endothelial genes which were specifically upregulated in the heart endothelium at baseline do not appear to be affected to the extent that the brain and lung endothelium were during LPS stimulation. In the heatmap (*Figure 9C*), a few genes such as Rho family GTPase 1 (*Rnd1*) and platelet glycoprotein (*Cd36*) undergo a robust change in expression during the time course. From our analysis, we found that the endothelial genes specific to the heart endothelium are much more abundant in the heart versus the other tissues. For example, caveolin 1 (*Cav1*) and vascular endothelial growth factor receptor 2 (*Kdr*) maintained a high expression level in the heart endothelial samples during the entire LPS time course whereas in the brain and lung endothelial samples, we see significantly lower expression.

We next focused of the organ-specific endothelial glycolysis signature to investigate the tissue-specific dynamics of glycolytic genes. The brain endothelial basal translatome upregulated the greatest number of glycolytic genes compared to the lung and heart endothelium. Interestingly, when we analyzed the time course of these brain endothelial specific glycolysis genes, we found that they maintain similar levels during the progression and resolution of inflammation (*Figure 9—figure supplement 1A*). There were only three glycolysis-related genes which were upregulated in the lung endothelium. When we analyzed these three genes over time, we found that two of them remained stable whereas 6-phosphofructo-2-kinase/fructose-2,6-biphosphatase 3 (*Pfkfb3*) was dynamic in all three tissues. Even though this glycolysis regulatory enzyme was specifically upregulated in the lung endothelium at baseline, we found that it was activated in all tissues during late inflammation/early repair and then returned to baseline levels (*Figure 9—figure supplement 1B*). In the heart endothelium, we found that the upregulated glycolytic genes were not modulated during the LPS injury and recovery (*Figure 9—figure supplement 1C*).

## Discussion

The endothelium which lines the entire vasculature evolves in a tissue-dependent manner during embryonic development to control organ development, homeostasis, and tissue regeneration (*Augustin and Koh, 2017*). Under normal physiological conditions, the endothelium maintains a quiescent interface between the blood and tissue. During inflammatory stimulation, the endothelium becomes actively responsible for controlling blood flow, vascular permeability, leukocyte infiltration, and tissue edema (*Pober and Sessa, 2015*). Understanding the organotypic endothelial heterogeneity that exists at baseline as well as during the transition from the normal state to the inflammatory state is essential for understanding endothelial plasticity in homeostasis and tissue-specific responses to inflammation (*Chaqour et al., 2018*; *Dejana et al., 2017*; *Krenning et al., 2016*; *Malinovskaya et al., 2016*).

The RiboTag strategy was originally applied to expression profiling of neurons and Sertoli cells (*Sanz et al., 2009*). Cell type specificity of the approach depends on the accuracy of the Cre driver that is combined with the $Rpl22^{HA}$ allele. This aspect is highlighted in our study and we revealed the precision of the inducible system for achieving endothelial specificity. Our results demonstrate that the RiboTag approach provides a useful method to identify distinct molecular gene expression signatures of tissue-specific endothelium. Performing high-throughput gene expression analysis on the translatome using the RiboTag approach enabled us to establish tissue-specific molecular signatures underlying in situ endothelial heterogeneity. During homeostasis, we found that the endothelial translatome in each organ is uniquely characterized by a signature adapted to the surrounding parenchymal tissue. The metabolic adaptation of the endothelium is less surprising as the endothelium plays a critical role in supplying nutrients to the host tissue (*Malinovskaya et al., 2016*; *Hamuro et al., 2016*). The upregulation of the glucose transporter 1 (*Slc2a1*) in brain ECs is consistent with the massive glucose consumption of the brain (*Schuenke et al., 2017*), whereas the upregulation of the fatty acid metabolism genes *Cd36* and *Fabp4* in the heart likely reflects the importance of fatty acids to meet the bioenergetic demands of cardiomyocytes (*Elmasri et al., 2009*; *Silverstein and Febbraio, 2009*). Similarly, the upregulation of immune and stress response

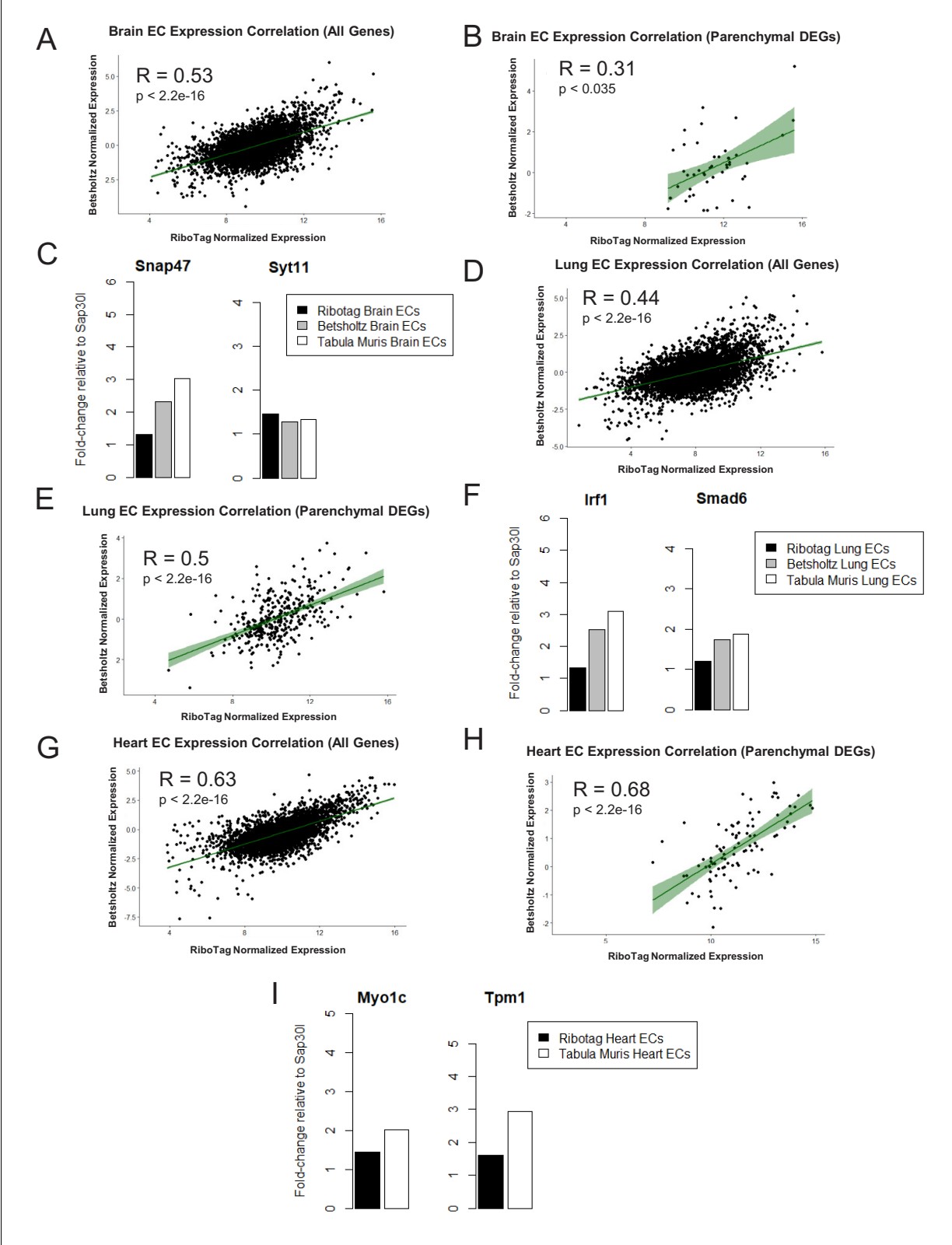

**Figure 6.** Expression Correlation Analysis between endothelial gene expression generated by RiboTag, Betsholtz, and Tabula Muris Spearman correlation scatter plots of average gene expression in RiboTag bulk RNA-Seq, Betsholtz scRNA-Seq (Smart-Seq2), and Tabula Muris scRNA-Seq (10x Genomics). (**A**) All genes detected in brain ECs. (**B**) Parenchymal (non-endothelial) genes detected in brain ECs. (**C**) All genes detected in lung ECs. (**D**)

*Figure 6 continued on next page*

*Figure 6 continued*

Parenchymal (non-endothelial) genes detected in lung ECs. (**E**) All genes detected in heart ECs. (**F**) Parenchymal (non-endothelial) genes detected in heart ECs.

The online version of this article includes the following figure supplement(s) for figure 6:

**Figure supplement 1.** Endothelial gene expression correlation analysis across three distinct datasets (RiboTag Endothelial Translatome, Betsholtz single-cell transcriptomics, Tabula Muris single-cell transcriptomics).

genes in the lung endothelium is expected due to the lung's continuous exposure to environmental stressors and pathogens contained in the inhaled air (*Al-Soudi et al., 2017*; *Kaparakis-Liaskos and Ferrero, 2015*).

However, the adaptation of the endothelium appears to extend far beyond the supply of metabolites and nutrients to the parenchyma. We surprisingly found that there exists a multidirectional molecular cross-talk of vessel wall cells with the cells of their microenvironment. In the brain endothelium, synapse organization and neurotransmitter transport genes such as *Glul* were highly enriched, which discloses the molecular mechanisms underlying how excitatory neurotransmitters such as glutamate can be transported among brain endothelial cells, neurons, and astrocytes (*Hawkins, 2009*). We also found that lung ECs expressed genes typically found in the lung epithelium such as Surfactant Protein C (*Spc*) and Mucin1 (*Muc1*), again indicative of a key interaction of the lung endothelium with the lung parenchymal epithelium. The upregulation of genes involved in cardiomyocyte contraction such as *Myl2* and *Ckmt2* again points to an unexpected adaptation of the cardiac endothelium to the surrounding cardiomyocytes, possibly suggesting a key role for the endothelium in modulating cardiac contractility (*Cai et al., 1998*; *Schnittler et al., 1990*).

Studying endothelial heterogeneity in response to the systemic inflammatory stress induced by LPS, we found that the endothelium in each tissue maintains a distinct organ-specific molecular identity. Brain and heart ECs express classical inflammatory adhesion molecules such as E-Selectin and P-Selectin, whereas lung ECs upregulate chemokines such as *Cxcl1* and *Cxcl9*. The gene expression shifts in the lung may also reflect the severe loss of lung endothelium recently observed during endotoxemia (*Merle et al., 2019*). The marked upregulation P-Selectin in the heart and brain is especially interesting because P-Selectin is a key mediator of thrombosis and platelet aggregation (*Merle et al., 2019*), and both the brain and heart are especially vulnerable to thrombotic events. During the later stage of inflammation at 24 hr, the inflammatory gene expression pathways across all tissues demonstrated significant upregulation of leukocyte migration and chemotaxis genes, suggesting that despite the persistent heterogenous signatures of the ECs in the respective organs, there is a broad shared program of inflammatory signaling pathways in response to systemic endotoxemia.

One of the requisites for targeted therapies is the need to deliver such agents to specific organs, thus underscoring the importance of leveraging organ-specific endothelial heterogeneity for such approaches. It has been suggested that vascular endothelial cells in different organs or disease states express specific markers, or 'zip codes' (*Folkman, 1999*), so that ligands directed against organ-specific vascular endothelial cell surface markers could be used to deliver effector molecules to specific vascular beds. To address this concept, we expanded our analysis by analyzing 1296 cell surface glycoproteins, including 136 G-protein coupled receptors and 75 membrane receptor tyrosine-protein kinases. This allowed us to establish EC surface markers that were specifically upregulated in in each vascular bed. Not only was this integrative analysis valuable for the establishment of EC 'zip codes' based on the organs they are derived from, but it may also provide insights about tissue-specific cell-cell contacts of ECs that allow them to interact with niche or parenchymal cells in each tissue (*Maoz et al., 2018*; *Zamani et al., 2018*).

Among the most intriguing findings of our study was the prominent 'parenchymal' signature of endothelial cells in each organ such as contractile genes in the cardiac endothelium and neurotransmitter transport or synaptic vesicle genes in the brain endothelium. A rank-based statistical analysis demonstrated that only selected genes of surrounding parenchymal cells were expressed in the endothelium of each organ. In the setting of a possible contamination, the most abundant genes expressed in the surrounding cells would also be the most abundant genes found in the cell of

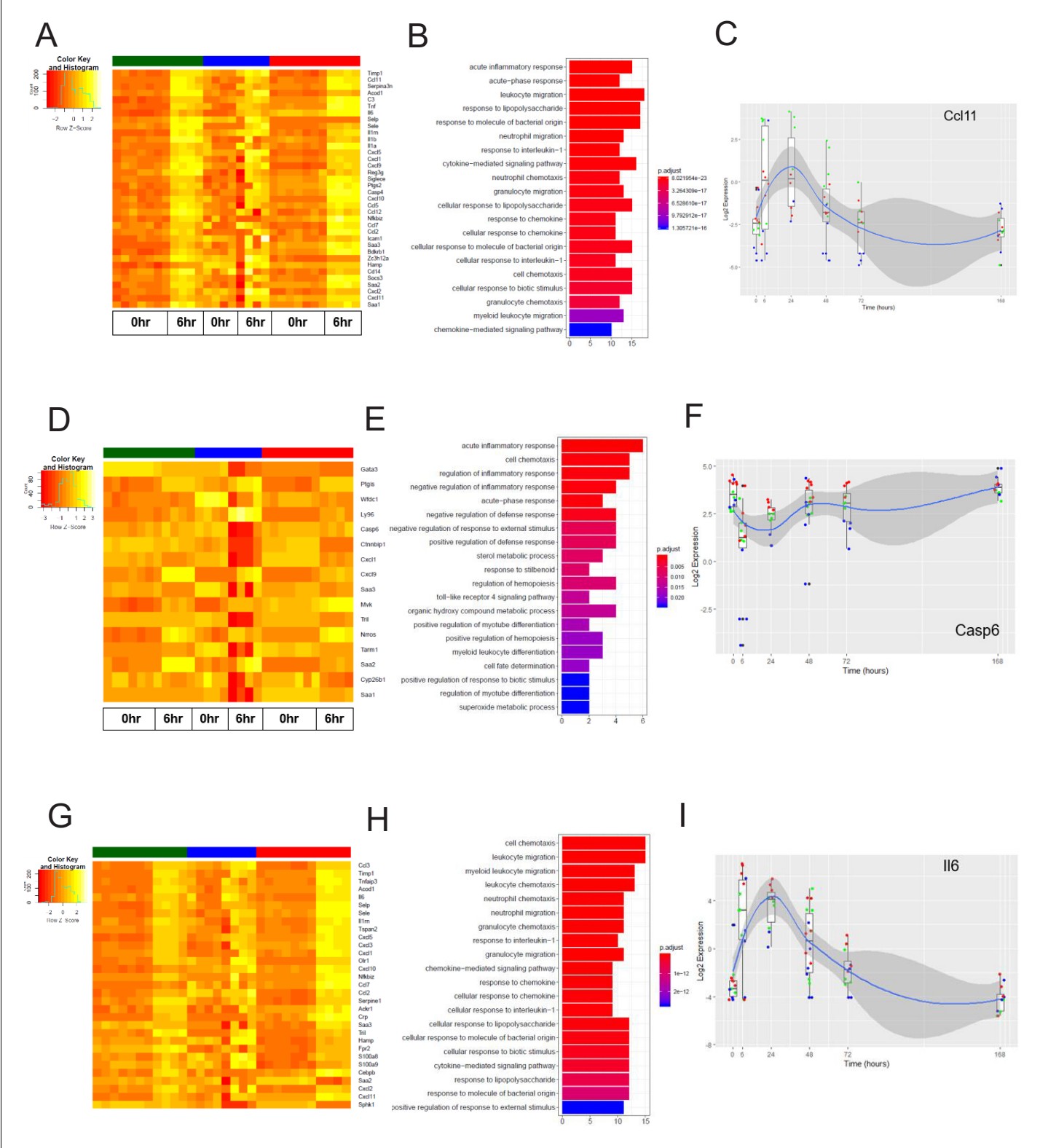

**Figure 7.** The early inflammation (6 hr) markers across organ-specific endothelial cells. (A) Heat map representation of differentially expressed genes identified by comparing brain ECs to lung and heart ECs at the 6 hr time point. The orange to yellow to white gradient represents increasing expression of the pathway with orange representing minimal expression while the white represents high expression of the pathway. (B) The GSEA results of enriched GO terms from RiboTag brain ECs at the 6 hr time point. (C) Tissue-specific kinetics of a specific RiboTag brain EC early inflammatory marker during the progression and resolution of inflammation. (D) Heat map representation of differentially expressed genes identified by

*Figure 7 continued on next page*

*Figure 7 continued*

comparing lung ECs to brain and heart ECs at the 6 hr time point. The orange to yellow to white gradient represents increasing expression of the pathway with orange representing minimal expression while the white represents high expression of the pathway. (E) The GSEA results of enriched GO terms from RiboTag lung ECs at the 6 hr time point. (F) Tissue-specific kinetics of a specific RiboTag lung EC early inflammatory marker during the progression and resolution of inflammation. (G) Heat map representation of differentially expressed genes identified by comparing heart ECs to brain and lung ECs at the 6 hr time point. The orange to yellow to white gradient represents increasing expression of the pathway with orange representing minimal expression while the white represents high expression of the pathway. (H) The GSEA results of enriched GO terms from RiboTag heart ECs at the 6 hr time point. (I) Tissue-specific kinetics of a specific RiboTag heart EC early inflammatory marker during the progression and resolution of inflammation.

interest. That the rank order of parenchymal genes abundance in the endothelium differed from that found in the parenchyma suggests tissue-specific programming and adaptation of the endothelium.

To further address the concern of possible mRNA contamination by neighboring cells in the RiboTag[EC] data, we systematically analyzed two independent endothelial single cell RNA-Seq datasets (*Vanlandewijck et al., 2018*; *Tabula Muris Consortium et al., 2018*), which can exclude contaminating tissue cells by examining the identity of each sequenced cell. We found that EC signature genes identified by our RiboTag[EC] approach such as the synaptic vesicle gene Snap47 and cardiac contractile gene Tropomyosin were also expressed in individual brain and heart ECs as identified by scRNA-Seq. Importantly, we found a substantial overlap of individual signature genes across our data and both scRNA-Seq datasets. Even though the approaches to obtain the data were so different, this is a remarkable degree of consilience. We used a genetic VE-cadherin-Cre to label endothelial ribosomes whereas the Tabula Muris scRNA-Seq dataset relied on mRNA markers of endothelial cells and Betsholtz dataset used Claudin5 lineage tracing combined with endothelial gene expression markers to identify individual ECs.

Although the bulk of scRNA-Seq tissue-specific genes were found in the Ribotag dataset, the converse was not true. Not all RiboTag[EC] signature genes were present in the single cell RNA-Seq datasets. We think this likely reflects the greater depth and sensitivity of Ribotag RNA-Seq because current single cell technologies are limited in their ability to detect the expression of individual genes in a given cell (*Bacher and Kendziorski, 2016*; *Zhu et al., 2018*; *Kharchenko et al., 2014*; *Lun et al., 2016*; *Vallejos et al., 2017*). Not all single ECs expressed parenchymal genes such as Tropomyosin or Snap47 but those expressing them did so at an even higher levels than what we found in the RiboTag[EC] data. The reason for this might be that RiboTag[EC] data represent an aggregate of all ECs in a tissue. It is therefore possible that the tissue adaptation of individual ECs may be most prominent in anatomically distinct ECs, for example those in close proximity to parenchymal cells such as neurons and astrocytes. Furthermore, if the expression of parenchymal gene signatures such as synaptic vesicle genes or cardiac contractile genes in the endothelium is dependent on environmental cues from neighboring cells or the extracellular matrix, the disassociation of the cells required for single cell RNA-seq may have further reduced mRNA levels of these genes (*Haimon et al., 2018*; *Rossner et al., 2006*; *Sugino et al., 2006*). Sequencing a larger number of individual ECs in these tissues may enable identification of additional EC subsets with the most prominent parenchymal signatures, and a single cell sequencing approach that preserves the anatomy of the tissue such as Slide-Seq (*Rodriques et al., 2019*) may also be useful to address the in situ transcriptomic signature.

Using the RiboTag model, we were able to characterize the endothelial translatome profile from distinct tissues. Our analysis uncovered a previously unrecognized degree of endothelial plasticity and adaptation to the parenchymal tissues, raising intriguing questions about the role that the endothelium plays in modulating parenchymal tissue function that likely go far beyond the classically ascribed roles of supplying oxygen, metabolites and solutes. Further studies such as endothelial-specific deletion of neurotransmitter transport or cardiac contractile genes will be required to establish the functional roles of these tissue-specific genes expressed in the endothelium of each organ. Understanding the biological significance of endothelial plasticity and adaptation to the parenchyma will be important in providing a fuller picture of endothelial function during homeostasis and stress in each tissue.

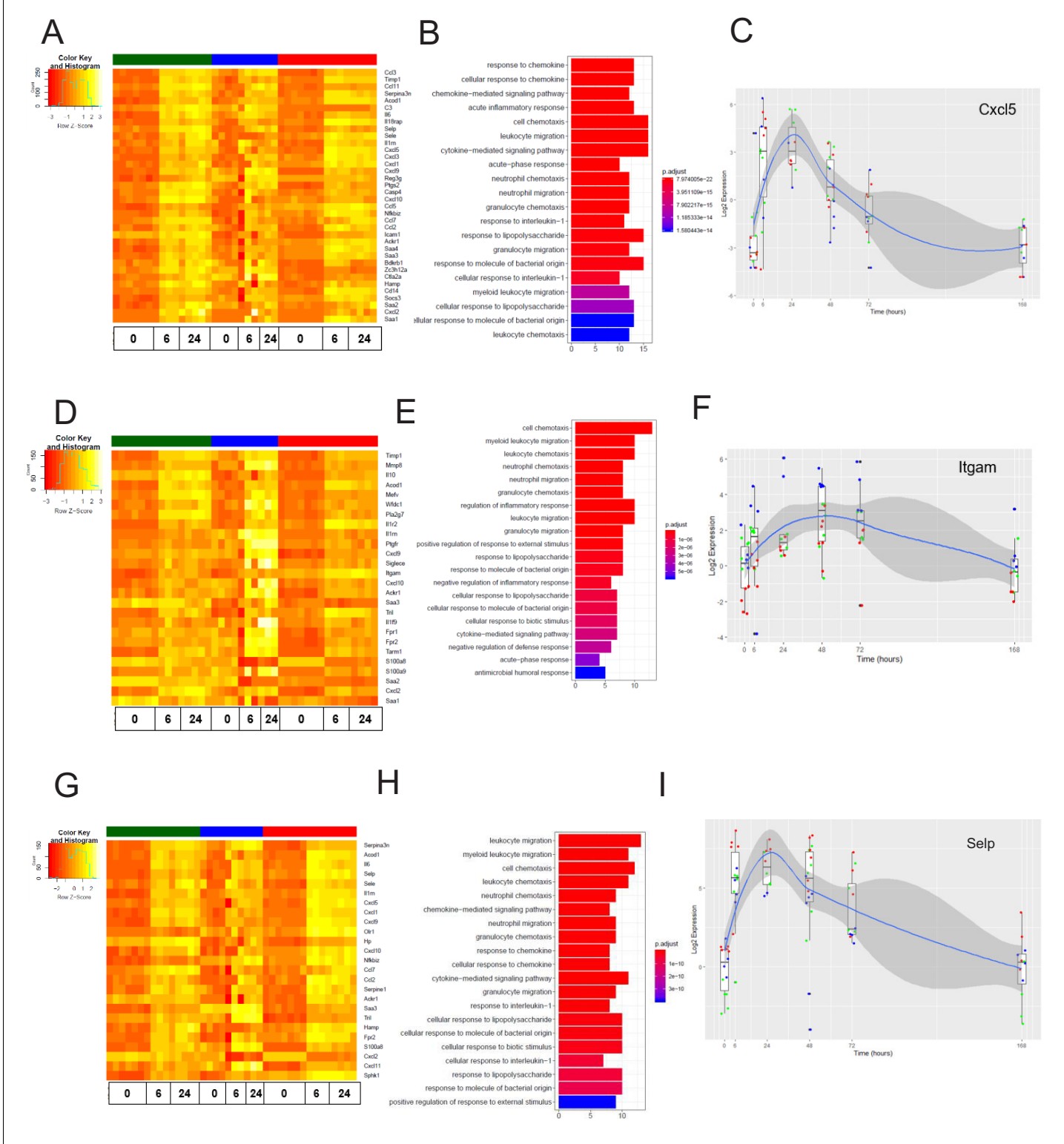

**Figure 8.** The late inflammation (24 hr) markers across organ-specific endothelial cells. Heat map representation of differentially expressed genes identified by comparing brain ECs to lung and heart ECs at the 24 hr time point. The orange to yellow to white gradient represents increasing expression of the pathway with orange representing minimal expression while the white represents high expression of the pathway. (B) The GSEA results of enriched GO terms from RiboTag brain ECs at the 24 hr time point. (C) Tissue-specific kinetics of a specific RiboTag brain EC late inflammatory marker during the progression and resolution of inflammation. (D) Heat map representation of differentially expressed genes identified by comparing lung ECs to brain and heart ECs at the 24 hr time point. The orange to yellow to white gradient represents increasing expression of the

*Figure 8 continued on next page*

*Figure 8 continued*

pathway with orange representing minimal expression while the white represents high expression of the pathway. (**E**) The GSEA results of enriched GO terms from RiboTag lung ECs at the 24 hr time point. (**F**) Tissue-specific kinetics of a specific RiboTag lung EC late inflammatory marker during the progression and resolution of inflammation. (**G**) Heat map representation of differentially expressed genes identified by comparing heart ECs to brain and lung ECs at the 24 hr time point. The orange to yellow to white gradient represents increasing expression of the pathway with orange representing minimal expression while the white represents high expression of the pathway. (**H**) The GSEA results of enriched GO terms from RiboTag heart ECs at the 24 hr time point. (**I**) Tissue-specific kinetics of a specific RiboTag heart EC late inflammatory marker during the progression and resolution of inflammation.

The online version of this article includes the following figure supplement(s) for figure 8:

**Figure supplement 1.** Markers of early (6 hr) and late (24 hr) LPS-induced inflammation in brain, lung, and heart ECs.

# Materials and methods

## Key resources table

| Reagent type (species) or resource | Designation | Source or reference | Identifiers | Additional information |
|---|---|---|---|---|
| Antibody | Anti-HA (Rabbit polyclonal) | Abcam | Cat#: Ab9110; RRID:AB_307019 | (1:133) |
| Antibody | Anti-IgG1(Mouse monoclonal) | Sigma | Cat#: M5284; RRID:AB_1163685 | (1:133) |
| Antibody | Anti-RPL22 (Rabbit polyclonal) | Invitrogen | Cat#: PA5-68320; RRID:AB_2692054 | (1:133) |
| Antibody | Anti-CD31 (Rat monoclonal) | BD Pharmingen | Cat#: 550274; RRID:AB_393571 | (1:25) |
| Antibody | Anti-RAGE (Rabbit polyclonal) | Abcam | Cat#: Ab3611; RRID:AB_303947 | (1:3200) |
| Antibody | Anti-PTN (Mouse monoclonal) | Santa Cruz Biotechnology | Cat#: sc-74443; RRID:AB_1128556 | (1:3200) |
| Antibody | Anti-AQP7 (Rabbit polyclonal) | Novus Biologicals | Cat#: NBP1-30862; RRID:AB_2258607 | (1:3200) |
| Antibody | Anti-rat (Donkey polyclonal) | Invitrogen | Cat#: A-21208; RRID:AB_141709 | (1:300) |
| Antibody | Anti-rabbit (Donkey polyclonal) | Invitrogen | Cat#: A-21207; RRID:AB_141637 | (1:300) |
| Antibody | Anti-mouse (Goat polyclonal) | Invitrogen | Cat#: A11032; RRID:AB_2534091 | (1:300) |
| Chemical compound, drug | Lipopolysaccharide (LPS) | Sigma-Aldrich | Cat#: L2630 | |
| Chemical compound, drug | collagenase A | Roche | Cat#:10103586001 | |
| Chemical compound, drug | red blood cell lysis buffer | Biolegend | Cat#: 420301 | |
| Chemical compound, drug | Dynabeads | Invitrogen | Cat#: 11035 | |
| Chemical compound, drug | Collagenase/ Dispase | Roche | Cat#: 11097113001 | |

*Continued on next page*

*Continued*

| Reagent type (species) or resource | Designation | Source or reference | Identifiers | Additional information |
|---|---|---|---|---|
| Chemical compound, drug | DNAse | Worthington Biochemical | Cat#: LK003170 | |
| Genetic reagent (*M. musculus*) | Mouse: *Cdh5*<sup>CreERT2/+</sup>; *Rpl22*<sup>HA/+</sup> | This paper | | Ref: Materials and methods – Experimental Animals |
| Genetic reagent (*M. musculus*) | *Rpl22*<sup>HA/+</sup> | Jackson Labs | JAX: 011029; RRID:IMSR_ JAX:011029 | |
| Genetic reagent (*M. musculus*) | *Cdh5*<sup>CreERT2/+</sup> | *Jeong et al., 2017*, *Sörensen et al., 2009* | | |
| Other | Myelin Removal Beads | Miltenyl Biotec | Cat#: 130-096-433 | |
| Other | LS columns | Miltenyl Biotec | Cat#: 130-042-401 | |
| Other | CD31 microbeads | Miltenyl Biotec | Cat#: 130-097-418; RRID:AB_2814657 | |
| Other | MACS BSA Stock Solution | Miltenyl Biotec | Cat#: 130-091-376 | |
| Other | autoMACS Rinsing Solution | Miltenyl Biotec | Cat#: 130-091-222 | |
| Other | MS columns | Miltenyl Biotec | Cat#: 130-042-201 | |
| Other | ProLong Gold mounting medium | Invitrogen | CA#: P36934 | |
| Software, algorithm | Zen software | ZEISS | | |
| Software, algorithm | STAR v. 2.4.2 | *Dobin et al., 2013* | | |
| Software, algorithm | HTSeq-count v. 0.6.1 | *Anders et al., 2015* | | |
| Software, algorithm | biomaRt package v. 2.26.1 | *Durinck et al., 2009* | | |
| Software, algorithm | ComBat | *Johnson et al., 2007* | | |
| Software, algorithm | limma | *Ritchie et al., 2015* | | |
| Software, algorithm | GSEA | *Subramanian et al., 2005* | | |
| Software, algorithm | Seurat | *Butler et al., 2018* | | |
| Software, algorithm | Tableau Public | Tableau Software | | |

## Experimental animals

RiboTag (*Rpl22*<sup>HA/+</sup>) mice were purchased from Jackson Labs. Endothelial-specific VE-cadherin-Cre mice were provided by Dr. Ralf Adams. We crossed the RiboTag mice (*Rpl22*<sup>HA/+</sup>) (*Sanz et al., 2009*) with the endothelial-specific VE-cadherin-Cre mice (*Jeong et al., 2017*; *Sörensen et al., 2009*) to generate RiboTag<sup>EC</sup> (*Cdh5*<sup>CreERT2/+</sup>; *Rpl22*<sup>HA/+</sup>) mice. Following tamoxifen-induced recombination at week 4, HA-tagged *Rpl22* was specifically expressed in endothelial cells. To investigate the mechanisms of organ-specific EC injury, repair, and regeneration we performed RNA-Seq

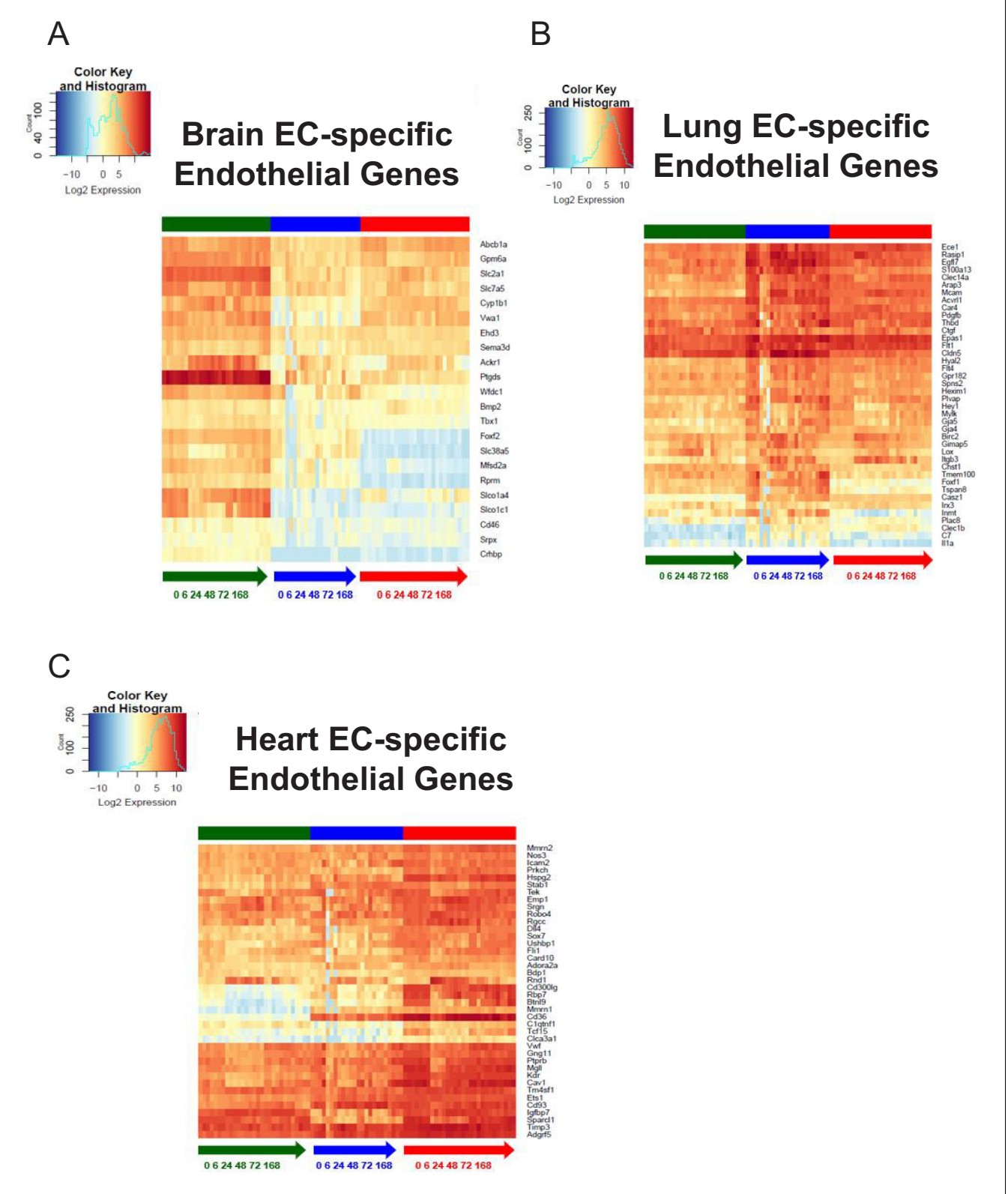

**Figure 9.** Organ-specific endothelial cells uniquely regulate endothelial genes during the progression and resolution of inflammation. (A–C) Time-series heat map of significantly upregulated endothelial genes at baseline in (A) brain ECs (B) lung ECs and (C) heart ECs. The blue to white to red gradient represents increasing expression of the pathway with blue representing minimal expression while the red represents high expression of the pathway. *Figure 9 continued on next page*

*Figure 9 continued*

The online version of this article includes the following figure supplement(s) for figure 9:

**Figure supplement 1.** Organ-specific endothelial cells uniquely regulate glycolysis genes during the progression and resolution of inflammation.

analysis of gene expression in ECs isolated at 6 hr, 24 hr, 48 hr, 72 hr, and 1 week post-LPS challenge (10 mg/kg LPS i.p., Sigma-Aldrich Cat#: L2630) with PBS-injected mice serving as controls.

The C57BL/6J mice were purchased from the Jackson Laboratory. All animal experiments were conducted in accordance with NIH guidelines for the Care and Use of Laboratory Animals and were approved by the IACUC of the University of Illinois (IACUC Protocol #19–014, IACUC Protocol #13–175 and IACUC Protocol #16–064).

## Isolation of mouse lung, heart and brain

After surgically opening the mouse chest, the brain, lung and heart were harvested after a one-time perfusion of 20 mL PBS through the left and right ventricular chamber.

## Ribosome immunoprecipitation (IP)

The tissue samples were extracted from RiboTag$^{EC}$ mice, flash-frozen in liquid nitrogen and then stored at $-80°C$. The samples were then homogenized on ice in ice-cold homogenization buffer (50 mM Tris, pH7.4, 100 mM KCl, 12 mM MgCl$_2$, 1% NP-40, 1 mM DTT, 1:100 protease inhibitor (Sigma), 200 units/mL RNasin (Promega) 1 mg/mL heparin and 0.1 mg/mL cycloheximide (Sigma) in RNase free DDW) 10% w/v with a Dounce homogenizer (Sigma) until the suspension was homogeneous. To remove cell debris, 1 mL of the homogenate was transferred to an Eppendorf tube and was centrifuged at 10,000xg and 4°C for 15 min. Supernatants were subsequently transferred to a fresh Eppendorf tube on ice, then 100 μL was removed for 'input' analysis and 3 μL (=3 μg) of anti-HA antibody (ab9110, Abcam) or 3 μL (=1 μg) of mouse monoclonal IgG1 antibody (Sigma, Cat# M5284) or 6 μL anti-RPL22 (Invitrogen Cat# PA5-68320) was added to the supernatant, followed by 1 hr of incubation with slow rotation in a cold room at 4°C. Meanwhile, Pierce Protein A/G Magnetic Beads (Thermo Fisher Scientific), 100 μL per sample, were equilibrated to homogenization buffer by washing three times. At the end of 1 hr of incubation with antibody, beads were added to each sample, followed by incubation 1 hr in cold room at 4°C. After that, samples were washed three times with high-salt buffer (50 mM Tris, 300 mM KCl, 12 mM MgCl2, 1% NP-40, 1 mM DTT, 1:200 protease inhibitor, 100 units/mL RNasin and 0.1 mg/mL cycloheximide in RNase free DDW), 5 min per wash in a cold room on a rotator. At the end of the washes, beads were magnetized, and excess buffer was removed, 350 μL Lysis Buffer was added to the beads and RNA was extracted with RNeasy plus Mini kit (Qiagen). RNA was eluted in 30 μL H$_2$O and taken for RNA-Sequencing.

## RNA-Sequencing

RNA quality and quantity were assessed using an Agilent Bio-analyzer. RNA-Seq libraries were prepared using Illumina mRNA TruSeq kits as protocolled by Illumina. Library quality and quantity were checked using an Agilent Bio-analyzer and the pool of libraries was sequenced using an Illumina HiSeq4000 and Illumina reagents.

## RNA-Sequencing data processing and batch correction

The sequenced reads from all samples were aligned to the mouse (mm10) reference genome with STAR v. 2.4.2 (*Dobin et al., 2013*), and the aligned reads were used to quantify mRNA expression by using HTSeq-count v. 0.6.1 (*Anders et al., 2015*). Gene symbols were mapped to the ENSEMBL features using the biomaRt package v. 2.26.1 (*Durinck et al., 2009*). Preliminary unsupervised analysis of normalized and processed profiles by principal component analysis (PCA) revealed separation into three major clusters. These clusters largely corresponded to the distribution of samples by sequencing batch. Consistent with the PCA plots, the distribution of samples by sequencing batch differed significantly but not by time point after inflammatory treatment or tissue type. To better harmonize profiles prior to analyses reported here, we normalized expression data of all samples using ComBat (*Johnson et al., 2007*). This correction ameliorated the separation by sequencing

batch without substantially affecting distributions by time point after inflammatory treatment and tissue type.

## Baseline tissue-specific gene signatures

We calculated the differential expression level of genes using a one versus others approach in order to identify signature genes which were upregulated for each tissue at baseline. For instance, to identify the genes significantly upregulated in brain ECs at baseline, we compared the 0 hr brain EC samples to 0 hr lung ECs and 0 hr heart ECs. We performed these analyses for all three tissues to identify baseline organ-specific EC signatures. We utilized the *limma* R package and applied the standard *limma* pipeline (*Ritchie et al., 2015*) to RNA-Seq data after voom transformation (*Law et al., 2014*). For each gene, the log fold-change (logFC) in expression level is derived from the *limma* analysis. Genes with FDR < 0.05 were identified as being differentially expressed. All upregulated genes for each tissue were plotted using the heatmap.2 function from the *gplots* v.3.0.1.1 (*Warnes, 2011*) R package. The top 10 significantly differentially expressed genes by logFC were listed.

## Baseline tissue-specific pathway analysis

To define the biological function associated with the molecular signature of the tissue-specific ECs, we specifically performed gene set enrichment analysis (GSEA) (*Subramanian et al., 2005*) on the genes which were significantly upregulated (logFC >1) in the tissue of interest. GSEA was performed on significantly upregulated genes ranked by their p-value using the clusterProfiler package (*Yu et al., 2012*) in R with gene ontology (GO) gene sets downloaded from the Molecular Signatures Database (MSigDB) (*Liberzon et al., 2015*). The top 20 most enriched GO terms were plotted.

## Baseline cell surface markers

Tissue-specific cell surface markers were identified by intersecting tissue-specific differentially expressed genes with predicted cell surface markers, as reported in the Cell Surface Protein Atlas (www.proteinatlas.org) (*Bausch-Fluck et al., 2015*). The top 10 significantly differentially expressed cell surface proteins by logFC were plotted.

## Isolation of lung ECs

The C57BL/6J mice mouse lungs were minced and digested with 3 mL collagenase A at 1 mg/mL in PBS (Roche, Cat#: 10103586001) at 37˚C water bath for 1 hr. Mixtures were titrated with #18 needles and then pipetted through a 40 μm disposable cell strainer. After centrifuging 500xg for 5 min and washing with 1x PBS, the isolated cells were treated with red blood cell lysis buffer (Biolegend, Cat#: 420301) for 5 min. After washing with 1x PBS twice, cells were incubated in suspension buffer ($Ca^{2+}$ and $Mg^{2+}$ free PBS, 0.5% BSA, 4.5 mg/mL D-glucose, and 2 mM EDTA) with 5 μg anti-CD31 antibody (BD Pharmingen, Cat#: 553370) at 4˚C for 60 min with gentle tilting and rotation. After washing, cells were then incubated in suspension buffer with pre-washed Dynabeads (20 μL beads in 1 mL buffer, Invitrogen Cat#: 11035) at 4˚C for 60 min with gentle tilting and rotation. After washing with 1x PBS three times using magnetic separation, lung ECs were dissociated from magnetic beads with trypsin.

## Isolation of brain ECs

The forebrains of C57BL/6J mice were micro dissected and minced in collagenase/dispase (Roche, Cat#: 11097113001) and DNAse (Worthington Biochemical Cat#: LK003170) and incubated for 1 hr at 37˚C. Myelin Removal Beads (Miltenyl Biotec, Cat#: 130-096-433) and LS columns (Miltenyl Biotec, Cat#: 130-042-401) were used. The resulting pellet after myelin removal contained microglia, astrocytes and endothelial cells. The endothelial cells were further enriched by using CD31 microbeads (Miltenyl Biotec, Cat#: 130-097-418).

## Isolation of heart ECs

Isolated C57BL/6J mice hearts were minced and digested with prewarmed Collagenase/Dispase mix (1 mg/mL) (Roche) at 37˚C for 30 min. 75 μL DNAse I per 10 mL cell suspension (1 mg/mL) was added and the suspension was incubated at 37˚C for 30 min. The digested tissue was filtered using

70 µm cell strainer followed by RBC lysis in RBC lysis buffer (Biolegend, Cat#: 420301) for 7 min at room temperature. The cell suspension was diluted with 10 mL of MACS buffer (Prepared in phosphate-buffered saline (PBS), pH 7.2, 0.5% bovine serum albumin (BSA), and 2 mM EDTA) by diluting MACS BSA Stock Solution (Cat#: 130-091-376) 1:20 with autoMACS Rinsing Solution (Cat#: 130-091-222)) and cells were passed through 40 µm cell strainer followed by centrifugation at 500xg for 5 min to pellet the cardiomyocytes. The supernatant containing endothelial cells was centrifuged at 800xg for 5 min to pellet down the ECs. The endothelial cell enriched pellet was resuspended in 500 µL of MACS buffer and the isolated cells were counted. Endothelial cells were further purified by using CD31 microbeads (Miltenyl Biotec, Cat#: 130-097-418) and Miltenyl Biotec MS columns (Miltenyl Biotec, Cat#: 130-042-201) through affinity chromatography according to the manufacturer's protocol.

## Preparation of cytospin slides from brain, lung and heart cells

The Thermo Shandon Cytospin three was used to generate Cytospin slides. Briefly, the Cytoslide with filter card were inserted into a Cytoclip. The Cytoclip was fastened and placed in a recess of the Cytospin rotor after sliding a Cytofunnel into it. The required volume of the cell suspension was pipetted into the Cytofunnel after cell counting and calculation. The Cytospin was centrifuged for 500 rpm for 5 min. The slide was fixed with 4% paraformaldehyde for 10 min and stored in 1x PBS at 4˚C.

## Immunofluorescence and confocal microscopy

The slides were permeabilized and blocked with 10% donkey serum, 2% BSA, 0.05% tween in PBS for 1 hr at room temperature. For lung cells, the slides were incubated with primary antibodies anti-CD31 (BD Pharmingen, Cat#: 550274, 1:25) and anti-RAGE (Abcam, Cat#: Ab3611, 1:3200) at 4˚C overnight. The brain ECs were incubated with primary antibodies anti-CD31 (BD Pharmingen, Cat#: 550274, 1:25) and anti-PTN (Santa Cruz Biotechnology, Cat#: sc-74443, 1:3200) at 4˚C overnight. For the heart samples, primary antibodies anti-AQP7 (Novus Biologicals, Cat#: NBP1-30862, 1:3200) and anti-CD31 (BD Pharmingen, Cat#: 550274, 1:25) were used and incubated at 4˚C overnight. The next day, slides were washed and incubated with the fluorescence-conjugated secondary antibody (AF488 donkey anti-rat 1:300, Invitrogen Cat#: A-21208; AF594 donkey anti-rabbit 1:300, Invitrogen Cat#: A-21207; AF594 goat anti-mouse 1:300, Invitrogen Cat#: A11032), followed by washing with 1x PBS. Cells were stained with DAPI and mounted on ProLong Gold mounting medium (Invitrogen, Cat#: P36934). Images were taken with a confocal microscope LSM880 (Zeiss) and analyzed by Zen software (Zeiss).

## Assessing baseline endothelial heterogeneity

Tissue-specific baseline gene expression heatmaps were generated for gene sets related to endothelial function including classical endothelial markers, glycolysis, fatty acid metabolism, and solute transport. The individual genes listed in the heatmaps contain the tissue-specific differentially expressed genes which overlapped with each of the respective gene sets.

The classical endothelial markers gene set contains 152 mouse endothelial cell markers downloaded from PanglaoDB (*Franzén et al., 2019*). The mouse glycolysis and fatty acid metabolism gene sets containing 67 and 52 genes respectively were downloaded from the Rat Genome Database (RGD) https://rgd.mcw.edu/ (*Shimoyama et al., 2015*). For the transport gene set, the solute carrier family including 423 membrane transport proteins located in the cell membrane were downloaded from the HUGO Gene Nomenclature Committee database (https://www.genenames.org/) (*Hediger et al., 2013*).

## Computational assessment of mRNA purity

Due to the endothelial cells being surrounded by other tissue-resident cell types, it is likely that the mRNA isolated from endothelial-specific RiboTag[EC] samples could contain non-endothelial mRNA. For this reason, we assessed the mRNA purity of RiboTag endothelial samples isolated from whole tissue by comparing the gene expression levels of the endothelial-specific RiboTag samples to the gene expression levels of mRNA from whole tissue. We compared endothelial-specific RiboTag[EC]

mRNA expression levels from brain, lung, and heart tissue to whole brain, lung, and heart mRNA expression levels.

We first acquired RNA-Seq data for whole brain, whole lung, and whole heart tissue from Array Express (*Athar et al., 2019*). The three whole brain samples and three whole lung samples were downloaded from accession number E-MTAB-6081, while the three whole heart samples were downloaded from accession number E-MTAB-6798. Raw mRNA counts were processed, and batch corrected in a cohort including the 0 hr RiboTag brain, lung, and heart endothelial mRNA counts. The preprocessing and batch correction were performed in the same manner as described above.

To identify whether mRNA of tissue-resident cells was isolated during the RiboTag EC mRNA isolation procedure, we calculated a Kendall's Tau rank coefficient between the most abundant genes in the RiboTag EC mRNA and whole tissue mRNA. The Kendall's Tau rank coefficient, ranging between −1 and 1, allowed us to test whether there was contamination of mRNA from the whole tissue in the RiboTag EC samples. As the coefficient approaches −1, the rank of most abundant genes differs in both sets of samples; while, as the coefficient approaches 1, the rank of most abundant genes becomes identical. Using this test, we were able to infer that if the rank of the most abundant genes in the RiboTag EC sample and the whole tissue is identical, there is contamination of non-endothelial mRNA in the RiboTag EC mRNA samples. All samples were compared to each other and heatmaps with Kendall's Tau rank coefficients were generated to visualize the results.

## Single-cell endothelial heterogeneity

To specifically analyze ECs at the single-cell level, we downloaded Tabula Muris data from https://github.com/czbiohub/tabula-muris and Betsholtz Lab data from NCBI Gene Expression Omnibus (GSE99235, GSE98816). We filtered out non-ECs from the Tabula Muris brain, lung, and heart data based on *Cd31* and *Cdh5* expression. We selected ECs from the Betsholtz Lab brain and lung data based on *Cd31* and *Cldn5* expression. All genes that were not detected in at least 10% of all single cells were discarded. For all further analyses we used 2585 cells expressing 6802 genes from the Tabula Muris dataset and 873 cells expressing 8116 genes from the Betsholtz Lab dataset. Data were log transformed for all downstream analyses. We analyzed the data utilizing the Seurat R package (https://github.com/satijalab/seurat; http://satijalab.org/seurat/) (*Butler et al., 2018*). PCA analysis of organ-specific ECs was performed in each dataset separately using the 'RunPCA' function of the Seurat package (*Butler et al., 2018*). Differential expression analysis for organ-specific endothelial cells was performed using a Wilcoxon rank-sum test as implemented in the 'FindAllMarkers' function of the Seurat package. GSEA was performed on significantly upregulated genes ranked by their p-value using the clusterProfiler package (*Yu et al., 2012*) in R with gene ontology (GO) gene sets downloaded from the Molecular Signatures Database (MSigDB) (*Liberzon et al., 2015*).

## Comparison of organ-specific endothelial translatome and endothelial single-cell transcriptomic data

Cross-platform comparisons between bulk RNA-Seq data and scRNA-Seq data required computing the fold change of each gene relative to a housekeeping gene. We calculated the relative fold change by dividing the expression value for every gene in every sample by an invariable housekeeping gene. We chose *Sap30l* as the housekeeping gene because it was invariable in all three datasets. By generating the fold change matrix in all three datasets, we were then able to use these values to compare relative abundances for genes of interest. We next calculated Spearman's correlation coefficients for all genes shared between the organ-specific endothelial translatome, Tabula Muris scRNA-Seq, and Betsholtz scRNA-Seq datasets, and then separately for all parenchymal (non-endothelial) genes.

## Tissue-specific endothelial kinetics following LPS-induced injury

To ascertain the kinetics of the tissue-specific endothelial signatures during inflammation we analyzed the time-series RNA-Seq data with the gene sets referenced above: classical endothelial markers, glycolysis, fatty acid metabolism, and transport. To visualize the tissue-specific dynamics for predominant endothelial functions, we plotted a heatmap which includes the tissue-specific differentially expressed genes for each gene set.

## Early and late tissue-specific inflammatory markers

To identify the inflammatory genes that were upregulated in the LPS 6 hr samples as compared to the baseline samples, we applied the standard *limma* pipeline (*Ritchie et al., 2015*) for genes in the 'inflammatory response' gene ontology term (GO:0006954). The analysis was carried out on the tissue specific LPS treated samples against the baseline tissue-specific sample. *Limma* statistically evaluates each inflammatory gene and returns the genes which show statistically significant change between the inflammatory time point and baseline. We applied this approach to the early inflammation time point, 6 hr, and the late inflammatory time point, 24 hr. Heatmaps were generated to visualize the tissue-specific inflammatory genes and their kinetics.

## Online endothelial translatome expression database ([www.rehmanlab.com/ribo](www.rehmanlab.com/ribo))

The endothelial translatome expression database is hosted on Amazon S3. The website was constructed using Angular 8.0, JavaScript, HTML5, and CSS. Barplots and heatmaps were generated for genes of interest using Tableau Public. The visualizations were integrated into the web application using the Tableau JavaScript API. RiboTag $log_2$ normalized baseline and inflammation time-course translatome expression data were uploaded to Tableu. The averages were computed using Tableau calculated fields. Tableau dashboards and workbooks were created to generate bar plots and heatmaps for online publishing.

## Acknowledgements

The studies were supported by NIH grants R01HL126516 (to JR), P01-HL60678 (to ABM and JR), T32-HL007829 (to ABM), R01-HL90152 (to JR and ABM) and AHA CDA grant 18CDA34110068 (to LZ). The $Cdh5^{CreERT2}$ mice were provided by Dr. Ralf Adams. We would like to thank Jing Du from Dr. Jan Kitajewski's group for providing advice on the isolation of RiboTag mRNA.

## Additional information

### Funding

| Funder | Grant reference number | Author |
|---|---|---|
| National Institutes of Health | R01HL126516 | Jalees Rehman |
| National Institutes of Health | P01-HL60678 | Asrar B Malik<br>Jalees Rehman |
| National Institutes of Health | T32-HL007829 | Asrar B Malik |
| National Institutes of Health | R01-HL90152 | Asrar B Malik<br>Jalees Rehman 9965552 |
| American Heart Association | 18CDA34110068 | Lianghui Zhang |

The funders had no role in study design, data collection and interpretation, or the decision to submit the work for publication.

### Author contributions

Ankit Jambusaria, Conceptualization, Data curation, Formal analysis, Validation, Visualization, Methodology; Zhigang Hong, Formal analysis, Methodology; Lianghui Zhang, Peter T Toth, Formal analysis, Validation, Methodology; Shubhi Srivastava, Asrar B Malik, Resources, Formal analysis, Supervision, Funding acquisition; Arundhati Jana, Jalees Rehman, Conceptualization, Resources, Data curation, Formal analysis, Supervision, Funding acquisition, Investigation, Methodology, Project administration; Yang Dai, Formal analysis, Supervision, Methodology

## Author ORCIDs

Ankit Jambusaria ⓘ https://orcid.org/0000-0003-3039-2300
Asrar B Malik ⓘ https://orcid.org/0000-0002-8205-7128
Jalees Rehman ⓘ https://orcid.org/0000-0002-2787-9292

## Ethics

Animal experimentation: All animal experiments were conducted in accordance with NIH guidelines for the Care and Use of Laboratory Animals and were performed in accordance with protocols approved by the Institutional Animal Care and Use Committees (IACUC) of the University of Illinois (protocol approval numbers 19-014, 13-175 and 16-064).

## Decision letter and Author response

Decision letter https://doi.org/10.7554/eLife.51413.sa1
Author response https://doi.org/10.7554/eLife.51413.sa2

# Additional files

## Supplementary files

• Supplementary file 1. Brain endothelial-specific gene list. The RiboTag brain endothelial signature genes are listed in rank order according to log fold-change (LogFC). The expression levels for all RiboTag EC samples are provided for each gene.

• Supplementary file 2. Lung endothelial-specific gene list. The RiboTag lung endothelial signature genes are listed in rank order according to log fold-change (LogFC). The expression levels for all RiboTag EC samples are provided for each gene.

• Supplementary file 3. Heart endothelial-specific gene list. The RiboTag heart endothelial signature genes are listed in rank order according to log fold-change (LogFC). The expression levels for all baseline RiboTag EC samples are provided for each gene.

• Supplementary file 4. Brain endothelial parenchymal signature. The RiboTag brain EC signature genes which are not found in the PanglaoDB list of endothelial cell marker genes.

• Supplementary file 5. Lung endothelial parenchymal signature. The RiboTag lung EC signature genes which are not found in the PanglaoDB list of endothelial cell marker genes.

• Supplementary file 6. Heart endothelial parenchymal signature. The RiboTag lung EC signature genes which are not found in the PanglaoDB list of endothelial cell marker genes.

• Supplementary file 7. Brain endothelial signature gene expression across translatome and single cell transcriptomes. The relative abundance for brain endothelial translatome signature genes in RiboTag brain EC translatome samples, Betsholtz brain endothelial single-cell transcriptomes, and Tabula Muris brain endothelial single-cell transcriptomes using a housekeeping gene, *Sap30l,* to calculate fold change.

• Supplementary file 8. Heart endothelial signature gene expression across translatome and single cell transcriptomes. The relative abundance for heart endothelial translatome signature genes in RiboTag heart EC translatome samples, Betsholtz heart endothelial single-cell transcriptomes, and Tabula Muris heart endothelial single-cell transcriptomes using a housekeeping gene, *Sap30l,* to calculate fold change.

• Supplementary file 9. Lung endothelial signature gene expression across translatome and single cell transcriptomes. The relative abundance for lung endothelial translatome signature genes in RiboTag lung EC translatome samples, Betsholtz lung endothelial single-cell transcriptomes, and Tabula Muris lung endothelial single-cell transcriptomes using a housekeeping gene, *Sap30l,* to calculate fold change.

• Transparent reporting form

## Data availability

RNA Sequencing data have been deposited in GEO under accession code GSE136848. We downloaded Tabula Muris data from https://github.com/czbiohub/tabula-muris and Betsholtz Lab data from NCBI Gene Expression Omnibus (GSE99235, GSE98816).

The following dataset was generated:

| Author(s) | Year | Dataset title | Dataset URL | Database and Identifier |
|---|---|---|---|---|
| Rehman J | 2019 | Endothelial Heterogeneity Across Distinct Vascular Beds During Homeostasis and Inflammation Using RiboTag Strategy | https://www.ncbi.nlm. nih.gov/geo/query/acc. cgi?acc=GSE136848 | NCBI Gene Expression Omnibus, GSE136848 |

The following previously published datasets were used:

| Author(s) | Year | Dataset title | Dataset URL | Database and Identifier |
|---|---|---|---|---|
| Vanlandewijck M, He L, Mäe M, Andrae J, Betsholtz C | 2017 | Single cell RNA-seq of mouse lung vascular transcriptomes | https://www.ncbi.nlm. nih.gov/geo/query/acc. cgi?acc=GSE99235 | NCBI Gene Expression Omnibus, GSE99235 |
| Vanlandewijck M, He L, Mäe M, Andrae J, Betsholtz C | 2017 | Single cell RNA-seq of mouse brain vascular transcriptomes | https://www.ncbi.nlm. nih.gov/geo/query/acc. cgi?acc=GSE98816 | NCBI Gene Expression Omnibus, GSE98816 |
| The Tabula Muris Consortium | 2018 | Tabula Muris: Transcriptomic characterization of 20 organs and tissues from *Mus musculus* at single cell resolution | https://www.ncbi.nlm. nih.gov/geo/query/acc. cgi?acc=GSE109774 | NCBI Gene Expression Omnibus, GSE109774 |

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
