## [Decision Letter]

**Acceptance summary:**

In this study, the authors have uncovered a complex organ-specific transcriptional pattern in the vasculature. Interestingly, endothelial cells were found to express genes that normally are associated with the parenchyma cells of the organ examined, speaking for a high degree of plasticity. In the revised version, the authors have addressed the remaining comments and criticisms. A key improvement is the inclusion an extensive comparison between their data and two published scRNA-seq data sets. The paper will be a useful resource for scientists interested in vascular biology.

**Decision letter after peer review:**

[Editors’ note: the authors submitted for reconsideration following the decision after peer review. What follows is the decision letter after the first round of review.]

Thank you for submitting your work entitled "Endothelial heterogeneity across distinct vascular beds during homeostasis and inflammation" for consideration by *eLife*. Your article has been reviewed by three peer reviewers, and the evaluation has been overseen by a Reviewing Editor and a Senior Editor. The reviewers have opted to remain anonymous.

Our decision has been reached after consultation between the reviewers, who were uniform in their view. Based on these discussions and the individual reviews below, we regret to inform you that your work will not be considered further for publication in *eLife*.

Though the reviewers recognized the importance of endothelial heterogeneity and the utility of the RiboTag approach, the potential cross contamination issue raised by reviewer 3 was a particular key concern for all three reviewers.

Reviewer #1:

The authors have used RiboTag purification of protein encoding RNA to characterize bulk gene expression in Cdh5-Cre expressing endothelial cells from the brain, heart and lung. The methodology is sound and the results nicely presented. The major question raised is simply what is the value of this study for the field? There are technical and conceptual limitations that argue the value is limited. First, while the study addresses general, inter-organ endothelial heterogeneity between heart, lung and brain, the approach of bulk RNAseq fails to address the high level of intra-organ heterogeneity, i.e. differences between endothelial cells in arteries, veins, capillaries, venules etc. Given the strength of single cell sequencing to address both of these issues simultaneously, the value of this approach seems limited. Second, the authors do not draw significant new biological insights from the data they have harvested. They discuss general points such as the top 10 genes expressed in the bulk endothelial populations from the different organs, but what this means for organ function or organ-specific vascular function is not pursued or validated in any specific manner. Finally, the inflammatory studies are limited to responses to LPS, a stimulus that is certainly inflammatory but laboratory-based and not easily translated to actual disease states. Overall, this study could provide some value as a resource if an excellent and highly accessible website were designed to do this (e.g. the Betsholtz site for brain endothelial gene expression (http://betsholtzlab.org/VascularSingleCells/database.html); otherwise its value does not seem adequately significant compared to existing single cell-based databases that already exist.

Reviewer #2:

The manuscript addresses the theme of heterogeneity of the tumor vasculature in lung, brain and heart. The authors sought to isolate cell-type specific ribosomes by Ribo-Tag methodology (Sanz et al., 2009), followed by RNA-seq. They uncovered a complex organ-specific transcriptional pattern in the vasculature. Interestingly, endothelial cells were found to express genes that normally are associated with the parenchyma cells of the organ examines, speaking for a high degree of plasticity. The authors also examined the transcriptional pattern following LPS administration and slow found interesting signatures.

The study seems technically well executed, addresses a timely and important topic, and some of the findings are interesting. Unfortunately, the manuscript, in the present form, is like many papers already published using single cell RNA-Seq, entirely descriptive and providing lists of genes without any functional data.

If the authors were able to provide functional validation some their signature and show that their analysis is informative in some disease models, the manuscript would be much more interesting.

Reviewer #3:

The authors have used genetically controlled RiboTag sequencing to analyze and compare the transcription profile of endothelial cells (ECs) from different organs. The RiboTag approach circumvents intrinsic limitations of other gene expression analysis strategies, such as alterations due to tissue dissociation or flow cytometric sorting, but it involves immunoprecipitation, which can result in noise caused by pull-down of unspecific RNAs especially when expression of the tagged ribosomal protein is confined to a small fraction of cells. In this context, it is certainly critical to assess whether ECs indeed share the expression of genes with cells from the surrounding organ. The authors suggest that this applies to metabolic signatures and transporters but, for the example of brain ECs, also to genes related to "neurotransmitter transport" or "synapse organization". Here, it is obviously puzzling that completely different cell types, namely neurons and ECs, would share very specialized transcripts relating to neuronal function. Fortunately, single cell RNA sequencing, which has its own limitations, can be used to confirm the observations made by the authors in an independent fashion and exclude cross-contamination effects seen in bulk sequencing or RiboTag data. Performing this test to the top RiboTag brain EC signature genes (Figure 3C) with the single cell database for heart and lung (http://betsholtzlab.org/VascularSingleCells/database.html; Vanlandewijck et al., 2018) shows that Ptgds is expressed by oligodendrocytes and fibroblasts but not ECs in brain. Atp1a2 is expressed by mural cells and fibroblasts but not by ECs. Ptn and Actb are widely expressed by many different cell types. Apoe, Apoe and Igf2 expression is absent from ECs, whereas Bsg and Spock2 show indeed substantial endothelial expression. Thus, even at the level of this superficial validation, EC expression can be only confirmed for 4 out of 10 genes.

As the Betzholtz database includes lung data, I have performed a similar test for the lung EC signature genes listed in Figure 4C. No or only very low endothelial expression can be seen for Sftpc (an epithelial marker), Ager, Wfdc2, Muc1, and Lyz1. Retnla and Hoxa5 are not covered by this dataset, while the other 3 genes show spurious endothelial expression.

Even if one takes into account that the authors have performed some computational tests (see Figure 2A) and tried to exclude that contamination is the cause of their organ-influenced EC signatures, the comparison with scRNA-seq data indicates that the opposite is the case. It has to be taken into account that the RiboTag is not protected against cross-contamination similar to immunoprecipitation experiments with proteins, which easily pull down highly abundant cytoskeletal proteins irrespective of the primary antibody used. During tissue lysis in the RiboTag protocol, certain transcripts from surrounding non-ECs may more easily end up as contamination than others, which might reflect differences in RNA structure, stability or association with RNA-binding proteins.

It is noted that the authors have used EC single cell data from the Tabula muris compendium in their analysis. Here they see some similarity with their own RiboTag data on the level of GO terms, but, unfortunately, the analysis stops here and individual signature genes are not validated.

Taken together, I am not convinced that the RiboTag data presented in the manuscript offers new and unexpected insights into organ and EC-specific gene expression. Instead, the purity of the RNA-seq data and its interpretation appear highly problematic so that I cannot recommend publication of this manuscript.

---

## [Author Response]

[Editors’ note: The authors appealed the original decision. What follows is the authors’ response to the first round of review.]

Reviewer #1:The authors have used RiboTag purification of protein encoding RNA to characterize bulk gene expression in Cdh5-Cre expressing endothelial cells from the brain, heart and lung. The methodology is sound and the results nicely presented. The major question raised is simply what is the value of this study for the field? There are technical and conceptual limitations that argue the value is limited. First, while the study addresses general, inter-organ endothelial heterogeneity between heart, lung and brain, the approach of bulk RNAseq fails to address the high level of intra-organ heterogeneity, i.e. differences between endothelial cells in arteries, veins, capillaries, venules etc. Given the strength of single cell sequencing to address both of these issues simultaneously, the value of this approach seems limited.

We agree that single cell sequencing can also be used to address inter-tissue and intra-tissue endothelial heterogeneity. In our studies, however, we were interested in identifying the endothelial translatome in vivo using the RiboTag approach driven by the endothelial specific VE-cadherin-Cre.

Therefore, the RiboTag provides a snapshot of the translatome in endothelial cells while residing in their physiological environment (i.e., in their context) without requiring tissue disaggregation and formation of single cell suspensions that disrupt cell function and gene expression (see van den Brink et al., Nature Methods 2017). The RiboTag method enriches for mRNAs undergoing translation whereas single cell RNA-seq analyzes total mRNA, including nuclear and cytosolic mRNA not undergoing translation. Therefore, the RiboTag approach is more suitable for identifying genes that are being actively translated into proteins whereas standard mRNA profiling (in bulk tissues or in single cells) identifies all mRNA, including mRNAs not undergoing translation. To clarify this point, we have revised the Introduction and included references highlighting the importance of translational regulation mechanisms which result in the preferential translation of certain mRNAs and which could not be ascertained by standard single cell RNA-seq.

Another difference between single-cell analysis and bulk RNA-Seq (whether total mRNA-Seq or ribosome-enriched mRNA-seq) is sequencing coverage. Our RiboTag RNA-seq approach detected 15,736 genes in the brain endothelium while endothelial single cell RNA-Seq analysis using the gold standard Smart-Seq approach such as that performed by the Betsholtz lab detected 8,116 genes in the brain endothelium. On the other hand, single cell RNA-Seq is suited for identifying functional endothelial subpopulations (such as arterial, venous, capillary subpopulations), which could not be identified by the RiboTag approach. Thus, RiboTag endothelial profiling serves as a complementary and equally important approach to single cell endothelial profiling strategies, each with its own strengths and weaknesses.

In response, we have now edited and expanded multiple paragraphs in the Introduction to clarify the scope of our study and to inform the readers of the respective advantages of using RiboTag bulk RNA-Seq as well as scRNA-Seq based approaches to derive complementary insights.

Second, the authors do not draw significant new biological insights from the data they have harvested. They discuss general points such as the top 10 genes expressed in the bulk endothelial populations from the different organs, but what this means for organ function or organ-specific vascular function is not pursued or validated in any specific manner.

We agree with the reviewer that a more comprehensive list of the differentially expressed genes beyond the top 10 genes shown in the main figures would be helpful to derive biological insights. In response, we have now included the full list of differentially expressed signature genes for each organ-specific endothelial tissue (Supplementary files 1-3), which will allow readers to use the data for biological studies of interest. We have also included a list of differentially expressed “parenchymal” (non-endothelial; i.e., not present in the current PanglaoDB database of endothelial cell genes) genes in each endothelial bed (Supplementary files 4-6).

Additionally, in Discussion (paragraph eight), we now emphasize the importance of performing endothelial-specific genetic deletion and mechanistic studies to ascertain the functional significance of the genes we identified in each vascular bed and how these endothelial genes may influence organ function.

Finally, the inflammatory studies are limited to responses to LPS, a stimulus that is certainly inflammatory but laboratory-based and not easily translated to actual disease states.

We agree with the reviewer that additional disease models to study the transcriptome response in each vascular bed will be of great value. However, we also want to highlight the translational relevance of the LPS model (which is central to understanding endotoxemia). Circulating LPS serves as a key mediator of disease in patients with bacteremia and sepsis (see Charbonney et al., 2016). Since our goal was to compare inflammatory transcriptomic responses in multiple vascular beds, we needed to induce systemic inflammation in a controlled manner. LPS administration induces the release of inflammatory mediators and immune cell activation, thus allowing us to establish the heterogeneity of endothelial responses resulting from systemic inflammatory stimulation. References emphasizing the resurgence of interest in clinically targeting LPS/endotoxemia are now cited in the Introduction.

Overall, this study could provide some value as a resource if an excellent and highly accessible website were designed to do this (e.g. the Betsholtz site for brain endothelial gene expression (http://betsholtzlab.org/VascularSingleCells/database.html); otherwise its value does not seem adequately significant compared to existing single cell-based databases that already exist.

We agree with the reviewer that our dataset would be a valuable resource for other researchers to query. Prior to the *eLife* submission we had made our data publicly available on NCBI GEO at the following link: https://www.ncbi.nlm.nih.gov/geo/query/acc.cgi?acc=GSE136848, but as the reviewer points out, researchers would benefit from a user-friendly visualization website that would not require downloading RNA-Seq datafiles from the public NCBI-GEO website.

In response, we have now generated a website http://www.rehmanlab.org/ribo that enables researchers to query and visualize the organ-specific endothelial translatome between distinct organ-specific endothelial tissues at baseline and during systemic inflammation.

We provide in Author response image 1 screen shots of the website which is accessible at http://www.rehmanlab.org/ribo. This will be especially of value to researchers who want to assess the heterogeneous expression of specific genes in the distinct vascular beds at baseline or dynamic changes in these genes following inflammatory injury. We agree with the reviewer that this significantly increases the utility and impact of our analysis.

**Author response image 1. respfig1:** Organ-specific endothelial translatome database.

Reviewer #2:The manuscript addresses the theme of heterogeneity of the tumor vasculature in lung, brain and heart. The authors sought to isolate cell-type specific ribosomes by Ribo-Tag methodology (Sanz et al., 2009), followed by RNA-seq. They uncovered a complex organ-specific transcriptional pattern in the vasculature. Interestingly, endothelial cells were found to express genes that normally are associated with the parenchyma cells of the organ examines, speaking for a high degree of plasticity. The authors also examined the transcriptional pattern following LPS administration and slow found interesting signatures.The study seems technically well executed, addresses a timely and important topic, and some of the findings are interesting. Unfortunately, the manuscript, in the present form, is like many papers already published using single cell RNA-Seq, entirely descriptive and providing lists of genes without any functional data.If the authors were able to provide functional validation some their signature and show that their analysis is informative in some disease models, the manuscript would be much more interesting.

We agree with the reviewer that functional studies that mechanistically establish the roles of the various key vascular bed specific signature genes and pathways we identified in disease would be very valuable. The goal of our study was to comprehensively establish the baseline heterogeneity of the endothelium as well as the organ-specific endothelial responses to inflammation. We chose a translationally relevant disease model – endotoxemia which induces a systemic inflammatory response, thus allowing us to study the inflammatory response in each vascular bed in a standardized manner. We found that even in the setting of a profound systemic inflammatory disease, the baseline heterogeneity of each vascular bed was maintained. However, each endothelial bed exhibited a highly characteristic inflammatory response. Understanding baseline and inflammatory heterogeneity of the vasculature would be critical for new tissue-specific mechanistic insights as well as targeted therapeutics aiming to modulate inflammation in specific vascular beds opposed to a “shot-gun” approach that would indiscriminately modulate vascular inflammation in all tissues.

In response, the revised Discussion clarifies the scope of our work as well as the importance of future functional studies.

Reviewer #3:The authors have used genetically controlled RiboTag sequencing to analyze and compare the transcription profile of endothelial cells (ECs) from different organs. The RiboTag approach circumvents intrinsic limitations of other gene expression analysis strategies, such as alterations due to tissue dissociation or flow cytometric sorting, but it involves immunoprecipitation, which can result in noise caused by pull-down of unspecific RNAs especially when expression of the tagged ribosomal protein is confined to a small fraction of cells. In this context, it is certainly critical to assess whether ECs indeed share the expression of genes with cells from the surrounding organ. The authors suggest that this applies to metabolic signatures and transporters but, for the example of brain ECs, also to genes related to "neurotransmitter transport" or "synapse organization". Here, it is obviously puzzling that completely different cell types, namely neurons and ECs, would share very specialized transcripts relating to neuronal function. Fortunately, single cell RNA sequencing, which has its own limitations, can be used to confirm the observations made by the authors in an independent fashion and exclude cross-contamination effects seen in bulk sequencing or RiboTag data. Performing this test to the top RiboTag brain EC signature genes (Figure 3C) with the single cell database for heart and lung (http://betsholtzlab.org/VascularSingleCells/database.html; Vanlandewijck et al., 2018) shows that Ptgds is expressed by oligodendrocytes and fibroblasts but not ECs in brain. Atp1a2 is expressed by mural cells and fibroblasts but not by ECs. Ptn and Actb are widely expressed by many different cell types. Apoe, Apoe and Igf2 expression is absent from ECs, whereas Bsg and Spock2 show indeed substantial endothelial expression. Thus, even at the level of this superficial validation, EC expression can be only confirmed for 4 out of 10 genes.As the Betzholtz database includes lung data, I have performed a similar test for the lung EC signature genes listed in Figure 4C. No or only very low endothelial expression can be seen for Sftpc (an epithelial marker), Ager, Wfdc2, Muc1, and Lyz1. Retnla and Hoxa5 are not covered by this dataset, while the other 3 genes show spurious endothelial expression.Even if one takes into account that the authors have performed some computational tests (see Figure 2A) and tried to exclude that contamination is the cause of their organ-influenced EC signatures, the comparison with scRNA-seq data indicates that the opposite is the case. It has to be taken into account that the RiboTag is not protected against cross-contamination similar to immunoprecipitation experiments with proteins, which easily pull down highly abundant cytoskeletal proteins irrespective of the primary antibody used. During tissue lysis in the RiboTag protocol, certain transcripts from surrounding non-ECs may more easily end up as contamination than others, which might reflect differences in RNA structure, stability or association with RNA-binding proteins.It is noted that the authors have used EC single cell data from the Tabula muris compendium in their analysis. Here they see some similarity with their own RiboTag data on the level of GO terms, but, unfortunately, the analysis stops here and individual signature genes are not validated.Taken together, I am not convinced that the RiboTag data presented in the manuscript offers new and unexpected insights into organ and EC-specific gene expression. Instead, the purity of the RNA-seq data and its interpretation appear highly problematic so that I cannot recommend publication of this manuscript.

We appreciate the in-depth comments and the reviewer’s comparisons of our data with the scRNA-seq data generated by the excellent work of the Betsholtz lab.

At the outset we realized this possible concern which is why we performed an extensive statistical analysis using the rank-based Kendall-tau test, which demonstrated that the gene expression rank order of “parenchymal genes” (such as cardiac contractile genes in the heart endothelium or neurotransmitter or synaptic vesicle genes in the brain endothelium) was distinct between the endothelium and parenchyma, thus reflecting tissue-specific endothelial plasticity instead of cross-contamination.

However, in direct response to the concern, we have carried out an in-depth analysis of single cell RNA-Seq data and comparison of single cell RNA-Seq data with our RiboTag data. This is important for establishing the complementary value of both approaches. The vascular biology field will greatly benefit from inclusive and comparative studies which utilize and are aware of the strengths of each approach. In response, we have performed the following extensive analysis to address the reviewer’s comments:

1) We have expanded our Introduction and Discussion sections to evaluate the parallel approaches available to study endothelial heterogeneity.

2) We have now included two new figures in the main manuscript (Figure 5, Figure 6, Figure 6—figure supplement 1) and have expanded the text of the manuscript in the Introduction, Results, and Discussion to address the concerns. We have also expanded our comparison of the signature genes identified by our VE-cadherin-Cre Ribotag approach to include both the Betsholtz single cell RNA Seq and the Tabula Muris single cell RNA Seq datasets. We to great satisfaction found a substantial overlap of individual genes across our data and the Betsholtz as well as Tabula Muris datasets. (Figure 5C-E, Results: subsection “Single-cell Endothelial Heterogeneity”)

Specifically, using the available scRNA-Seq endothelial datasets for all three organs we had analyzed in our RiboTag experiments, we first intersected the differentially expressed genes for each organ-specific endothelial population to identify the shared “signature” genes and pathways that were present with all three platforms (our RiboTag approach for brain, lung and heart ECs; the Betsholtz single cell studies which were performed on brain and heart ECs; the Tabula Muris single cell studies which were performed on brain, lung and heart ECs). We found that in the Tabula Muris and Betsholtz single cell RNA-Seq analysis of brain ECs, synapse organization, neurotransmitter transport, and regulation of ion transmembrane transport were also significantly enriched (Figure 5C), consistent with the brain EC-specific enrichment of neuronal signaling pathways identified by our RiboTagEC analysis. We also found that in lung ECs, the genes specifically upregulated in the Tabula Muris and Betsholtz single cell analyses were enriched for T cell activation, TGFβ signaling, and antigen processing and presentation (Figure 5D), consistent with what we had found using the RiboTagEC approach. Furthermore, we found that the heart EC marker genes enriched in the Tabula Muris single cell heart ECs (there were no heart ECs in the Betsholtz dataset) were involved in processes such as cardiac muscle contraction, myofibril assembly and proliferation (Figure 5E). By demonstrating that these signatures are present in three different and complementary approaches to tissue-specific EC profiling, we believe that it has substantially increased the robustness of our conclusions that ECs in distinct vascular beds express genes reflecting biological processes that define neighboring cells such as synaptic vesicle transport genes in brain ECs and cardiac contractility genes in heart ECs.

3) In response to the reviewer’s request to provide a broader, more comprehensive overview of the overlap, we further analyzed whether or not the top 50 genes identified by RiboTag were detected in single cell samples. For the brain, 80% of the top 50 brain EC specific genes in Tabula Muris single cell EC analysis were also present in the list of top 50 Ribotag brain EC genes and more than 50% of the top Betsholtz brain EC specific genes were present in the top 50 list of our Ribotag brain EC genes. Nearly 60% of the top lung endothelial specific genes in the Betsholtz data set were also found in the top 50 list of lung endothelial-specific genes in the RiboTag model. Nearly 50% of the top lung endothelial specific genes in the Tabula Muris data set were also found in the top 50 list of lung endothelial-specific genes in the RiboTag model. This is a remarkable overlap despite the analytic approaches being so different; we used a genetic VE-cadherin Cre to label EC ribosomes whereas Tabula Muris relied on mRNA markers of ECs such as CD31 and VE-cadherin and the Betsholtz group used a combination of Claudin5 endothelial lineage tracing and mRNA expression profiles to identify endothelial cells.(Figure 5F-G, Results: subsection “Single-cell Endothelial Heterogeneity”)

4) In response to the reviewer’s request to assess whether the parenchymal signatures (Supplementary files 4-6) we identified in the endothelial translatome were only driven by lowly abundant transcripts, we performed Spearman correlation analysis to compare organ-matched RiboTag bulk RNA-Seq data with scRNA-Seq data generated by the Betsholtz Lab and the Tabula Muris Compendium (Figure 6, Figure 6—figure supplement 1). One of the challenges for comparing bulk RNA-Seq and single cell RNA-Seq data is the profound difference in sequencing depth and coverage because single cell RNA-Seq by its very nature only identifies a comparatively low number of transcripts in a given cell. We therefore identified stable housekeeping genes that could be used as a reference point for comparing these three datasets (RiboTag, Betsholtz, Tabula Muris). This allowed us to compare the expression levels for each gene relative to the housekeeping gene. In each dataset, we first determined the fold change for all genes using the housekeeping gene, Sap30l (Supplementary files 7-9). Using these fold change values, we calculated correlation coefficients between the brain endothelial translatome and the single cell brain ECs from the Betsholtz and Tabula Muris datasets. We found that for all genes detected in the brain endothelium, the correlation between RiboTag and Betsholtz was 0.53 (Figure 6A).

We then specifically tested whether the “parenchymal” signature genes in the brain endothelium were correlated with the Betsholtz and Tabula Muris individual brain ECs. The correlation of the parenchymal gene expression between RiboTag brain EC samples and Betsholtz brain ECs was 0.31 (Figure 6B), indicating a correlation even in these parenchymal genes. Importantly, the abundance of the genes in the Betsholtz dataset (Figure 6B, y-axis) shows that they are not exclusively low abundance genes but instead comparable to the gene expression level distribution in the whole transcriptome (Figure 6A, y-axis)

Importantly, we now directly compare the expression levels of selected brain EC parenchymal genes such as the synaptosome associated protein 47 (Snap47) and synaptotagmin 11 (Syt11) which we found were expressed at similar or higher levels in the Betsholtz single cell brain ECs than in the RiboTag brain EC samples (Figure 6C).

We performed an identical analysis for the lung and heart endothelium (Figure 6D-I, Figure 6—figure supplement 1) and found that similar correlation values ranging between 0.37 to 0.68. Of note, the heart endothelial gene expression was the most correlated organ across the RiboTag EC and single cell heart EC platforms (Figure 6G-H). In the heart endothelium, we also found that individual genes representing the parenchymal signature such as the cardiomyocyte contractile protein Tropomyosin (Tpm1) are expressed at similar or higher levels in the single cell samples (Figure 6I).

5) As a resource, we are also now including the complete parenchymal signature for each tissue (Supplementary files 4-6) and the relative expression levels for all tissue-specific differentially expressed genes in each of the three datasets (Supplementary files 7-9) which directly address the reviewer’s concern about the abundance of the genes.